# Online Learning of Whittle Indices for Restless Bandits with Non-Stationary Transition Kernels

## Abstract

We study optimal resource allocation in restless multi-armed bandits (RMABs) under unknown and non-stationary dynamics. Solving RMABs optimally is PSPACE-hard even with full knowledge of model parameters, and while the Whittle index policy offers asymptotic optimality with low computational cost, it requires access to stationary transition kernels - an unrealistic assumption in many applications. To address this challenge, we propose a Sliding-Window Online Whittle (SW-Whittle) policy that remains computationally efficient while adapting to time-varying kernels. Our algorithm achieves a dynamic regret of $\tilde{O}(T^{2/3}\tilde{V}^{1/3} + T^{4/5})$ for large RMABs, where $T$ is the number of episodes and $\tilde{V}$ is the total variation distance between consecutive transition kernels. Importantly, we handle the challenging case where the variation budget is unknown in advance by combining a Bandit-over-Bandit framework with our sliding-window design. Window lengths are tuned online as a function of the estimated variation, while Whittle indices are computed via an upper-confidence-bound of the estimated transition kernels and a bilinear optimization routine. Numerical experiments demonstrate that our algorithm consistently outperforms baselines, achieving the lowest cumulative regret across a range of non-stationary environments.

## 1 Introduction

Many sequential decision-making problems can be modeled as restless multi-armed bandits (RMABs). A decision maker needs to choose $M$ out of $N$ arms to activate at each time-slot. Each arm is modeled as a Markov decision process, and evolves stochastically according to two different transition kernels, depending on whether the arm is activated or not. At the beginning of each time-slot, the decision maker picks a subset of arms to be activated. The activated arms evolve according to their active Markov transition kernels, while the rest of the arms evolve according to their passive Markov transition kernels. At the end of the time-slot, the decision maker receives rewards from each arm, where rewards are functions of the current state and the action.

RMABs have a long history in resource allocation and operations research literature, starting with Whittle's seminal work Whittle (1988) in the 1980s. Over the past four decades, RMABs have been used to model and optimize resource allocation problems in a wide variety of domains such as wireless scheduling Borkar et al. (2017); Kadota et al. (2018); Tripathi & Modiano (2024); Shisher et al. (2024); Kadota et al. (2019), machine monitoring and control Liu et al. (2011); Ruiz-Hernández et al. (2020); Dahiya et al. (2022), server scheduling Dusonchet & Hongler (2003), recommendation systems Meshram et al. (2017; 2018), and health care Villar et al. (2015); Bhattacharya (2018); Lee et al. (2019); Mate et al. (2020); Behari et al. (2024). In all of these applications, transition kernels can be unknown and non-stationary, i.e., the laws governing the evolution of states can drift over time. For example, consider a load balancing problem, where jobs arrive into a datacenter and a decision-maker assigns jobs to servers via a load balancer. The time required to finish a job at any server depends on its current load and how the load evolves over time. This evolution is typically random and time-varying since there are multiple load balancers and job streams contributing to the load at any given server within a large datacenter. Deciding which server to pick can then be formulated as an RMAB, but with non-stationary transition kernels.

When the transition kernels of a RMAB are unknown and non-stationary, the problem of finding the Whittle index becomes an online/reinforcement learning problem. Many papers designed algorithms for MDPs and MAB with unknown and non-stationary transition kernels and analyzed dynamic regret for MDPs and MAB Ortner et al. (2020); Cheung et al. (2020); Marin Moreno et al. (2024); Wei et al. (2023); Wei & Luo (2021). However, algorithm designed in these prior works can not be applied to RMAB due its special structures: The passive arms (arms that are not activated) continue to evolve stochastically. Because of the special structure and the combinatorial action space, even when the transition kernels are known, developing an optimal policy for RMABs is PSPACE-hard Papadimitriou & Tsitsiklis (1994). Whittle's seminal work Whittle (1988) introduced a heuristic policy for RMAB problem, known as the Whittle index policy. This policy relies on establishing a special mathematical property called *indexability* for each arm and then deriving functions called index functions that map states to how valuable it would be to activate an arm at that state. Running the policy simply requires activating the $M$ bandits with the highest Whittle indices out of the $N$ bandits at each decision time. To compute Whittle index, the problem is decomposed to multiple single-arm MDPs after a Lagrangian relaxation technique. Then, Whittle index is computed using the solution of the multiple MDPs. The Whittle index achieves asymptotic optimality, if the RMAB is indexable and has a global attractor point Weber & Weiss (1990); Verloop (2016); Gast et al. (2023; 2021). Most prior works in utilizing Whittle index-based policy focus on known and stationary transition kernels Dance & Silander (2015); Tripathi & Modiano (2024); Shisher et al. (2024); Le Ny et al. (2008); Meshram et al. (2018).

Applying traditional online learning and reinforcement learning policies Ortner et al. (2020); Cheung et al. (2020); Marin Moreno et al. (2024); Wei et al. (2023); Wei & Luo (2021) naively to RMAB with unknown and non-stationary transition kernels may lead to inefficient learning performance and to exponential regret bounds. This necessitates combining Whittle with online learning methods. In this direction, a recent work Wang et al. (2023)) designed a Whittle index-based policy called *UCWhittle* for unknown but stationary transition kernels. Although techniques exist for adapting reinforcement learning algorithms to non-stationary environments Wei & Luo (2021), they are not directly compatible with the recently proposed UCWhittle policy Wang et al. (2023). This incompatibility arises from the unique structure of RMABs and the specific method used to compute the Whittle index via Lagrangian relaxation.

In addition, it is common in many applications to have prior knowledge regarding the sparsity of transition kernels for some parts of the state space. For example, consider a wireless scheduling problem which aims to maximize information freshness in selecting which users (arms) to schedule. In this case, the state can be modeled using Age of Information (AoI) Kaul et al. (2012); Sun et al. (2016) – a widely used metric for quantifying information freshness. Then, the AoI of an arm increases by one if the arm is not scheduled for transmission. Conversely, if the arm is scheduled, its AoI resets to one with the success probability of the transmission. Thus, the AoI will never increase by 2 or decrease to a value other than 1.

In this paper, we pose the following research question: ***Can we develop a Whittle index-based online algorithm for RMABs with non-stationary transition kernels?***

**Contributions:** The main contributions of our paper can be summarized as follows:

- **Algorithm Design.** The challenge for designing online learning algorithms for RMABs is to incorporate the computationally efficient class of policies such as Whittle index policy into an adaptive process. We design a sliding window-based online Whittle index policy for non-stationary RMABs (see Algorithm 1). We model non-stationarity of transition kernels of arm $n$ by using a total variation budget $V_n$ which is an upper bound of the sum of the total variational distance $\tilde{V}_n$. To estimate the budget $V_n$, we utilize a Bandit-over-Bandit approach Cheung et al. (2022); Wei et al. (2023), in which $V_n$ is selected from a finite set of possible values. Based on the estimated $V_n$, the Whittle index is predicted by using a sliding window and upper confidence bound approaches. Moreover, our algorithm takes into account the sparsity of the transition kernels. This significantly simplifies the complexity of optimization and helps to predict the transition kernels accurately.

- **Dynamic Regret Analysis.** We rigorously characterize an upper bound on the dynamic regret of our algorithm. Our paper is the first to provide dynamic regret for the online learning of Whittle index under non-stationary environments. It is difficult to analyze dynamic regret of an online policy under non-stationary environments. It is even more difficult for RMABs. Wang et al. (2023) overcame this challenge by analyzing the regret for stationary environment using the Lagrangian

relaxed form of the problem and its solution. In this paper, we extend the regret analysis to (i) non-stationary environments and (ii) to a stronger version of regret by directly analyzing the main problem and its solution, instead of the Lagrangian form. Our policy can achieve dynamic regret of $\tilde{O}(T^{2/3}\tilde{V}^{1/3} + T^{4/5})$ for large system size when RMAB is indexable and has a global attractor point (see Theorem 3 & Remark 1).

- **Simulation Results.** Our simulation results (see Table 1 & Fig. 1) show that our algorithm achieves lower regret in practice compared with the UCWhittle policy Wang et al. (2023), WIQL policy Biswas et al. (2021), and a uniformly randomized policy Kadota et al. (2018) baselines.

## 2 RELATED WORK

**Offline Whittle Index Policy for RMABs:** Whittle's seminal work Whittle (1988) introduced a heuristic policy for the infinite-horizon RMAB problem, known as the Whittle index policy. Motivated by Whittle's work, many subsequent works have applied the Whittle index framework to different resource allocation problems Dance & Silander (2015); Tripathi & Modiano (2024); Shisher et al. (2024); Le Ny et al. (2008); Meshram et al. (2018); Kadota et al. (2018; 2019); Ornee & Sun (2023) by modeling them as RMABs.

**Online Learning of Whittle Index:** Multiple works Avrachenkov & Borkar (2022); Fu et al. (2019); Biswas et al. (2021) have proposed Q-learning algorithms to compute Whittle Index. Authors in Nakhleh et al. (2021) proposed NeurWIN and Nakhleh et al. (2022) proposed DeepTOP to compute Whittle index by using neural networks. These prior works did not provide any regret guarantees for their policy. In Tripathi & Modiano (2021), the authors develop an online Whittle algorithm with static regret guarantees compared to the best fixed Whittle index policy. Wang et al. (2023) is the first to provide the regret analysis for the online learning of Whittle index with unknown transition kernels. However, Wang et al. (2023) consider a stationary environment. In Wang et al. (2023), authors analyzed regret of UCWhittle by using Lagrangian relaxed form of the RMAB problem. We, in this paper, propose an online learning of Whittle index for *non-stationary* transition dynamics, with provable regret bounds. To the best of our knowledge, this is the first work to provide dynamic regret analysis of an online Whittle index-based policy for RMABs with non-stationary transitions.

## 3 PROBLEM SETTING

We consider an episodic RMAB problem with $N$ arms and an unknown non-stationary environment. Each arm $n \in [N]$ is associated with a unichain MDP denoted by a tuple $(\mathcal{S}, \mathcal{A}, P_{n,t}, r_n)$ at every episode $t$, where the state space $\mathcal{S}$ is finite, $\mathcal{A} = \{0, 1\}$ is a set of binary actions, $P_{n,t} : \mathcal{S} \times \mathcal{A} \times \mathcal{S} \mapsto [0, 1]$ is the transition kernel of arm $n$ with $P_{n,t}(s'|s, a)$ being the probability of transitioning to state $s'$ from state $s$ by taking action $a$ in episode $t$, and $r_n(s, a)$ is the reward function for arm $n$ when the current state is $s$ and the action $a$ is taken. The total number of episodes is $T$ and each episode itself consists of $H$ time slots. We consider that the transition kernels $P_{n,t}$ are unknown and non-stationary, i.e., $P_{n,t}$ can change across episodes $t \in [T]$.

A decision maker (DM) determines what action to apply to each arm at a decision time $h \in [H]$ of an episode $t \in [T]$ under the instantaneous activation constraint that at most $M$ arms can be activated. The action taken by the DM in episode $t$ is described by a deterministic policy $\pi_t : \mathcal{S}^N \mapsto \mathcal{A}^N$ which maps a given state $(s_1, s_2, \ldots, s_N) \in \mathcal{S}^N$ to an action $(a_1, a_2, \ldots, a_N) \in \mathcal{A}^N$. The corresponding expected discounted sum of rewards in episode $t$ is given by

$$R_t\left(\pi_t, (P_{n,t})_{n=1}^N\right) := \mathbb{E}\left[\sum_{h=1}^H \sum_{n=1}^N \gamma^{h-1} r_n(s_{n,h,t}, a_{n,h,t}) \middle| \pi_t, (P_{n,t})_{n=1}^N\right], \quad (1)$$

where $s_{n,h,t} \in \mathcal{S}$ is the state of arm $n$ at time $h$ of episode $t$, $a_{n,h,t} \in \mathcal{A}$ is the action taken by the DM for arm $n$ at decision time slot $h$ of episode $t$, and $\gamma$ is the discount factor. The DM aims to maximize the total expected sum reward across all episodes, subject to arm activation constraints, i.e.,

$$\max_{\pi_t \in \Pi} R_t\left(\pi_t, (P_{n,t})_{n=1}^N\right); \quad \text{s.t.} \sum_{n=1}^N a_{n,h,t} \leq M, \forall h \in [H], \forall t \in [T] \quad (2)$$

where $\Pi$ is the set of all causal policy $\pi_t : \mathcal{S}^N \mapsto \{0, 1\}^N$.

### 3.1 Lagrangian Relaxation

Because the main problem described in equation 2 is intractable, we relax the per-time slot constraint and use the Lagrangian defined below:

$$\mathbb{E}\left[\sum_{h=1}^{H}\sum_{n=1}^{N}\gamma^{h-1}\bigg(r_n(s_{n,h,t},a_{n,h,t})-\lambda a_{n,h,t}\bigg)\bigg|\pi_t,(P_{n,t})_{n=1}^{N}\right], \tag{3}$$

where $\lambda \geq 0$ is a Lagrangian penalty that is interpreted as the cost to pay for activation.

The Lagrangian problem described in equation 3 enables us to decompose the combinatorial decision problem equation 2 into a set of $N$ independent Markov decision process for each arm:

$$U\left(\pi_{n,t},P_{n,t},\lambda\right) = \max_{\pi_{n,t}\in\Pi_n}\mathbb{E}\left[\sum_{h=1}^{H}\gamma^{h-1}\bigg(r_n(s_{n,h,t},a_{n,h,t})-\lambda a_{n,h,t}\bigg)\bigg|\pi_{n,t},P_{n,t}\right], \tag{4}$$

where $\pi_{n,t}^*$ is the optimal solution that maximizes equation 4 from the set of all causal policies $\Pi_n$

### 3.2 Whittle Index Policy

Given $\lambda$, we denote by $\phi_n(\lambda)$ the set of states for which it is optimal not to activate the arm. The set $\phi_n(\lambda)$ is given by $\phi_n(\lambda) := \{s \in \mathcal{S} : Q_{n,\lambda,t}(s,0) > Q_{n,\lambda,t}(s,1)\}$, where the action value function $Q_{n,\lambda,t}(s,a)$ associated with Bellman optimality equation for equation 4 is

$$Q_{n,\lambda,t}(s,a) = r_n(s,a) - \lambda a + \gamma\sum_{s'\in\mathcal{S}}P_{n,t}(s'|s,a)V_{n,\lambda,t}(s') \tag{5}$$

and the value function $V_{n,\lambda,t}(s)$ associated with Bellman optimality equation for equation 4 is

$$V_{n,\lambda,t}(s) = \max_{a\in\mathcal{A}}Q_{n,\lambda,t}(s,a). \tag{6}$$

Intuitively, as the Lagrangian cost $\lambda$ increases, it is less likely the optimal policy activates arm $n$ in a given state. Hence, the set $\phi_n(\lambda)$ would increase monotonically.

**Definition 1 (Indexability)** *An arm is said to be indexable if the set $\phi_n(\lambda)$ increases monotonically as $\lambda$ increases from 0 to $\infty$. A restless bandit problem is said to be indexable if all arms are indexable.*

**Definition 2 (Whittle Index)** *Given indexablity and transition kernel $P_{n,t}$, the Whittle index $W_{n,t}(s;P_{n,t})$ of arm $n$ at state $s \in \mathcal{S}$ in episode $t$ is defined as:*

$$W_{n,t}(s;P_{n,t}) := \inf\{\lambda : Q_{n,\lambda,t}(s,0) = Q_{n,\lambda,t}(s,1)\}. \tag{7}$$

The Whittle index $W_{n,t}(s;P_{n,t})$ represents the minimum activation cost at which activating arm $n$ in state $s$ at episode $t$ is equally optimal to not activating it.

**Whittle Index Policy** activates at most $M$ arms out of $N$ arms with highest Whittle indices. However, as we can observe from equation 7, we can compute Whittle index if we know the transition kernel $P_{n,t}$ of every episode $t \in [T]$. Next, we model how transition kernels change over every episode.

### 3.3 The Transition Kernel Model

**Non-Stationarity:** In this section, we model the transition kernels for our non-stationary RMAB setting. We assume that the transition kernels $P_{n,t}$ may drift at varying rates across different arms $n \in [N]$ with the constraint that the total variation distance between transition kernels of two consecutive episodes is bounded from above by

$$\max_{(s,a)\in\mathcal{S}\times A}\sum_{s'\in\mathcal{S}}\left|P_{n,t}(s'|s,a) - P_{n,t-1}(s'|s,a)\right| \leq \frac{V_n}{T}, \tag{8}$$

---

**Algorithm 1:** Sliding Window-based Online Whittle Policy

---

**input :** State Space $\mathcal{S}$, Action Space $\mathcal{A}$, Reward Function $r_n(s,a)$ for all $(s,a)$ and arms $n$

1 DM initializes a Lagrange cost $\lambda^{(1)}$

2 **for** *every episode* $t = 1, 2, \ldots, T$ **do**

3     DM predicts variation budget $V_n$ for all $n \in [N]$

4     DM decides window size $w_n = \lceil (T/V_n)^{2/3} \rceil$ for all $n \in [N]$

5     Arm $n$ starts with state $s_{n,0}$

6     DM predicts $\tilde{P}_{n,t}$ for all arm $n \in [N]$ using equation 12 with $\lambda^{(t)}$.

7     DM computes Whittle Index $W_{n,t}(s) \forall s \in \mathcal{S}, n \in [N]$ with $\tilde{P}_{n,t}$ using equation 7.

8     **for** $h = 1, 2, \ldots, H$ **do**

9        DM activates $M$ arms (i.e., action=1) with highest Whittle Indices $W_{n,t}(s_{n,h,t})$.

10       All arms $n$ moves to the next state $s_{n,h+1,t} \sim P_{n,t}(\cdot | s_{n,h,t}, a_{n,h,t})$

11       DM observes states and updates counts $C_{t,w_n}^{(n)}(s_{n,h+1,t}, s_{n,h,t}, a_{n,h,t})$

12     Update $\lambda^{(t+1)} =$ M-th highest Whittle Index

---

where $V_n$ is the total variation budget across the entire $T$ episodes. The total variation budget $V_n$ represents the total non-stationarity in arm $n$ across the entire horizon, and is a standard quantity used for analzing dynamic regret in online learning literature Ortner et al. (2020); Cheung et al. (2020).

**Sparsity:** In many applications, the probability transition kernels are sparse - meaning that many state transitions are not possible under certain actions. To model this we introduce $\mathcal{S}_0(s,a)$ as the set of all states $s' \in \mathcal{S}$ such that the probability to transit from state $s \in \mathcal{S}$ to state $s' \in \mathcal{S}$ given action $a \in \mathcal{A}$ is always 0, i.e.,

$$\mathcal{S}_0(s,a) = \{s' \in \mathcal{S} : P_{n,t}(s'|s,a) = 0, \forall t\}.$$

The sets $\mathcal{S}_0(s,a)$ for all $(s,a) \in \mathcal{S} \times \mathcal{A}$ represents the sparsity of transition kernels for arm $n$. Our proposed algorithm can utilize this sparsity to reduce the complexity of the algorithm as described in Appendix A.6. Further, if we know sparsity (even approximately), we can use this information to learn faster by reducing exploration for certain transitions. Even in the absence of any sparsity, our results hold and the proposed algorithm are able to guarantee sublinear dynamic regret.

The DM is assumed to know the parameter $\mathcal{S}_0(s,a)$ for all $(s,a) \in \mathcal{S} \times \mathcal{A}$. In next section, we develop our Algorithm 1 that (i) learns the total variation budget and the probability transition kernels, and (ii) uses them to compute the Whittle Index to pick approximately optimal policies in each episode. In the next section, we discuss how we obtain our online algorithm.

## 4 SLIDING WINDOW-BASED ONLINE WHITTLE POLICY

To compute the Whittle index, we need to know transition kernels. In practice, transition kernels $P_{n,t}$ are unknown and non-stationary. In this section, we present Algorithm 1, an online approach for RMABs which adapts to unknown and non-stationary transition kernels.

Our *sliding window-based online Whittle policy*, provided in Algorithm 1, is motivated by the *UCWhittle* approach proposed in Wang et al. (2023). However, the *UCWhittle* policy is designed for static settings and does not handle time-varying transition kernels. This motivates the two main technical innovations in our policy. First, we employ a sliding window method that tracks transition kernels of the past $w_n$ episodes instead of all past episodes. The parameter $w_n$ is decided based on the total variation budget $V_n$. Second, we change the confidence bound provided in Wang et al. (2023). In designing the new confidence bound, we add a prediction horizon $w_n V_n / T$. We also discuss in Section 4.3 how we estimate the total variation budget $V_n$.

### 4.1 SLIDING WINDOW AND CONFIDENCE BOUNDS

At each episode $t$ and for each arm $n$, we maintain variables $C_{t,w}^{(n)}(s', a, s)$, which count the number of transitions from state $s$ to the state $s'$ via the action $a$ observed within the past $w$ episodes, i.e.

the sliding window. By using the counts for past $w$ episodes, we compute the empirical transition probabilities

$$\hat{P}_{n,t,w}(s'|s,a) := \frac{C^{(n)}_{t,w}(s',a,s)}{C^{(n)}_{t,w}(s,a)}, \tag{9}$$

where we define $C^{(n)}_{t,w}(s,a) := \max\left\{\sum_{s'\in\mathcal{S}} C^{(n)}_{t,w}(s',a,s), 1\right\}$. Using the upper confidence bound approach, we consider the following confidence radius

$$d^n_t(s,a) = \sqrt{\frac{2|\mathcal{S}|\log(2|\mathcal{S}||\mathcal{A}|NT/\eta)}{C^{(n)}_{t,w}(s,a)}} + \frac{w_n V_n}{T}, \tag{10}$$

where $\eta > 0$ is a design parameter. Notice that the term $\frac{w_n V_n}{T}$ in the confidence radius $d^n_t(s,a)$ measures how far the transition kernels could have drifted over a window of $w_n$ episodes.

Equipped with these definitions, the ball $B^{(n)}_t$ of the possible values for transition probabilities $P_{n,t}(s'|s,a)$ at any episode $t$ can be characterized as follows

$$B^{(n)}_t = \left\{ P_{n,t} : \sum_{s'\in\mathcal{S}} \left| P_{n,t}(s'|s,a) - \hat{P}_{n,t,w_i}(s'|s,a) \right| \le d^{(n)}_t(s,a), \right.$$

$$\left. P_{n,t}(s'|s,a) = 0, \forall s' \in \mathcal{S}_0(s,a), \sum_{s'\in\mathcal{S}} P_{n,t}(s'|s,a) = 1, \forall (s,a) \in \mathcal{S} \times \mathcal{A} \right\}. \tag{11}$$

We will show later that the true transition kernel lies within this high-dimensional ball with high probability in each episode.

## 4.2 ONLINE WHITTLE INDICES

Similar to Wang et al. (2023), we predict the transition probabilities in an optimistic approach. We select the optimistic transition probability $\tilde{P}_{n,t}$ for each arm $n$ that maximizes the value function within the confidence bound. The optimization problem for predicting the transition probability $\tilde{P}_{n,t}$ is given by

$$\max_{P_{n,t}\in B^{(n)}_t} V_{n,\lambda,t}(s), \text{ s.t. } V_{n,\lambda,t}(s) = \max_{a\in\mathcal{A}} Q_{n,\lambda,t}(s,a), \tag{12}$$

$$Q_{n,\lambda,t}(s,a) = r_n(s,a) - \lambda a + \sum_{s'} P_{n,t}(s'|s,a) V_{n,\lambda,t}(s') \tag{13}$$

As a result of the maximization procedure of equation 12, the true value function is upper bounded by the value function under the predicted transition kernel provided that the confidence bound in equation 11 holds. This upper bound value function will later allow us to prove regret bounds. Using the predicted transition kernel $\tilde{P}_{n,t}$, we compute $W_{n,t}(s : \tilde{P}_{n,t})$, the Whittle index of state $s \in \mathcal{S}$ for arm $n$ as defined in equation 7. Finally, we update Lagrange multiplier $\lambda^{t+1}$ as the $M$-th highest Whittle index at time slot $H$ of episode $t$. Detailed analysis of the computational complexity due to the kernel maximization problem equation 12 is discussed in Appendix A.6.

## 4.3 ESTIMATION OF UNKNOWN VARIATION BUDGET

In the above discussions, the total variation budget $V_n$ is assumed to be known. Now, we discuss how to adapt with the unknown variation budget $V_n$. We adopt the Bandit over Bandit approach Cheung et al. (2022); Wei et al. (2023) for the estimating the variation budget $V_n$. In this estimation approach, we solve another bandit problem to select $V_n$ from a finite set of possible budget values based on the history by using EXP3 algorithm Auer et al. (2002). Modified EXP3 algorithm for our problem is provided in Algorithm 2.

Next, we discuss how we can create a set of possible budget values. First, we get the maximum value for the variation budget as $V_{n,max} = 2T$. This holds because

$$\max_{(s,a) \in \mathcal{S} \times A} \sum_{s' \in \mathcal{S}} \left| P_{n,t}(s'|s,a) - P_{n,t-1}(s'|s,a) \right| \leq 2. \tag{14}$$

By using $V_{n,max}$, we can now define the set of qunatized drift values as $\{V_{n,max}, V_{n,max} - V_{n,max}/J_n, V_{n,max} - 2V_{n,max}/J_n, \ldots, V_{n,max}/J_n\}$, where $J_n$ is the number of quantization levels. This approach of quantizing and approximately estimating drift values is novel within online learning literature. We will show in Theorem 1 and Theorem 2 that the number of levels $J_n$ affects the dynamic regret (more levels means more accurate tracking of $V_n$ but also slower learning in the Bandit-over-Bandit approach).

## 5 REGRET ANALYSIS

We determine the regret of the policy $\pi_t$ in episode $t$ by subtracting the performance of our policy from the performance of the optimal policy (both under the true unknown transition kernel $P_{n,t}$). The cumulative dynamic regret in $T$ episodes is given by

$$\text{Reg}(T) = \sum_{t=1}^{T} \left( R_t \left( \pi_t^*, (P_{n,t})_{n=1}^{N} \right) - R_t \left( \pi_t, (P_{n,t})_{n=1}^{N} \right) \right)$$

$$\leq \sum_{t=1}^{T} \left( \sum_{n=1}^{N} U \left( \pi_{n,t}^*, P_{n,t}, \lambda \right) - R_t \left( \pi_t, (P_{n,t})_{n=1}^{N} \right) \right)$$

$$= \underbrace{\sum_{t=1}^{T} \left( \sum_{n=1}^{N} U \left( \pi_{n,t}^*, P_{n,t}, \lambda \right) - \sum_{n=1}^{N} U \left( \pi_{n,t}, P_{n,t}, \lambda \right) \right)}_{\text{Term1}}$$

$$+ \underbrace{\sum_{t=1}^{T} \left( \sum_{n=1}^{N} U \left( \pi_{n,t}, P_{n,t}, \lambda \right) - R_t \left( \pi_t, (P_{n,t})_{n=1}^{N} \right) \right)}_{\text{Term2}}, \tag{15}$$

where $\pi_{n,t}^*$ is the optimal policy of the problem defined in equation 4 associated with transition kernel $P_{n,t}$ and $\pi_{n,t}$ is the optimal policy of the the problem defined in equation 4 associated with transition kernel $\tilde{P}_{n,t}$. The first inequality holds because relaxed Lagrangian upper bounds the main problem.

Term1 is regret on the Lagrangian relaxed problem. To analyze the performance of Whittle index policy, Wang et al. (2023) only used the Lagrangian relaxed problem to assess the performance of an online learning algorithm. We consider a stronger version of regret definition by considering Term2 compared to Wang et al. (2023). Term2 is the performance difference between the Lagrangian problem and the original problem with the Whittle index policy derived using the solution of the Lagrangian problem.

First, we analyze Term1. Note that sublinear dynamic regret is usually challenging to establish in online learning literature, since we are comparing to a dynamic optimal policy that knows the entire sequence of transition kernels Besbes et al. (2015; 2019). We will show that our approach has sublinear dynamic regret, as long as the transition kernels don't vary too quickly.

To create a regret bound, we first need to establish how good our estimates of the time-varying transition kernel are. To do so, we will bound the probability that the true kernel is outside the high-dimensional ball $B_t^{(n)}$ introduced in equation 11. Lemma 1 describes the result in detail.

**Lemma 1** *Given $\eta \geq 0$, the probability that the true kernel $P_{n,t}$ lies within the high-dimensional Ball $B_t^{(n)}$ (described by eq. 11) is greater than or equal to $1 - \eta$, i.e., $\Pr(P_{n,t} \in B_t^{(n)}, \forall n, \forall t) \geq 1 - \eta$.*

Lemma 1 implies that for *every* episode $t$, we can provide a confidence region in which true transition kernel will lie with high probability. A detailed proof of Lemma 1 is provided in Appendix A.1.

Next, using this result, Theorem 1 characterizes the upper bound for $\text{Term}1$.

**Theorem 1** *With probability $1 - \eta$, the cumulative dynamic regret of Algorithm 1 satisfies:*

$$\text{Term}1 \leq \sum_{t=1}^{T} O\bigg( \sum_{n=1}^{N} 2|\mathcal{S}|G_{t,n}(w_n) \bigg) + \sum_{n=1}^{N} O\bigg( w_n(\tilde{V}_n + 2T/J_n)H \bigg)$$

$$+ \sum_{n=1}^{N} O\bigg( \sqrt{J_n \log(J_n)T} \bigg), \tag{16}$$

*where $G_{t,n}(w) = \max_{(s,a)\in\mathcal{S}\times\mathcal{A}} g_{t,n}(s,a,w)$ and the function $g_{t,n}(s,a,w) = \mathbb{E}_{P_{n,t},\pi_{n,t}}[\alpha_t^{(n)}(s,a)/\sqrt{C_{t,w}^{(n)}(s,a)}]$ is non-increasing in $w$, where $\alpha_t^{(n)}(s,a)$ is a random variable that denotes the number of visit at $(s,a)$ in episode $t$, $J_n$ is the number of elements in the set of quantized drift values for arm $n$ and $\tilde{V}_n$ is the actual total variation measure, given by*

$$\tilde{V}_n = \sum_{t=1}^{T} \max_{(s,a)\in\mathcal{S}\times A} \sum_{s'\in\mathcal{S}} \bigg| P_{n,t}(s'|s,a) - P_{n,t-1}(s'|s,a) \bigg|. \tag{17}$$

**Proof.** See Appendix A.2.

The regret bound for $\text{Term}1$ involves three error components: First term is a transition kernels learning error that decreases with the window size $w_n$; Second term is a transition kernels prediction error that increases with both $w_n$ and the variation budget $\tilde{V}_n$ but decreases with $J_n$; Third term is a variation budget learning error that increases with $J_n$. The following theorem simplifies this bound.

**Theorem 2** *If there exists a positive probability to visit every $(s,a) \in \mathcal{S} \times \mathcal{A}$ at least once in any episode $t \in [T]$ for all arms $n \in [N]$ and $w_n = \lceil(1/\epsilon_n)^{2/3}\rceil$, then with probability $1 - \eta$, we have*

$$\text{Term}1 \leq \sum_{n=1}^{N} \tilde{O}(T^{2/3}(\tilde{V}_n + 2T/J_n)^{1/3}) + \tilde{O}(\sqrt{TJ_n}).$$

**Proof.** See Appendix A.3.

To develop our final regret bound, we introduce $h(N)$ which is a function of the number of arms $N$

$$h(N) = \sum_{n=1}^{N} U\left(\pi_{n,t}, \tilde{P}_{n,t}, \lambda\right) - R_t\left(\pi_t, (\tilde{P}_{n,t})_{n=1}^{N}\right), \tag{18}$$

where $R_t(\pi_t, (P_{n,t})_{n=1}^{N})$ and $U(\pi_n, P_{n,t}, \lambda)$ are defined in equation 2 and equation 4, respectively. The function $h(N)$ represents the gap between the performance of the Lagrangian problem under its optimal solution and the main problem under Whittle index Policy. In $h(N)$, both the optimal solution of Lagrangian problem and the Whittle index Policy are designed and evaluated under same transition kernel $\tilde{P}_{n,t}$. Hence, $h(N)$ does not reflect the learning errors or regret, rather it measures the inherent optimality gap of the Whittle index policy even if we know transition kernels accurately.

Now, we are ready to present the upper bound of the cumulative regret term $\text{Reg}(T)$.

**Theorem 3** *Under the conditions of Theorem 2, if $J_n = O(T^{3/5})$, with probability $1 - \eta$, we have*

$$\text{Reg}(T) \leq \tilde{O}(T^{2/3}\tilde{V}^{1/3} + T^{4/5}) + h(N)T, \tag{19}$$

*where $\tilde{V} = \max_n \tilde{V}_n$.*

**Proof.** See Appendix A.4.

**Remark 1** *According to Theorem 3, the upper bound of $\text{Reg}(T)$ is given by $\tilde{O}(T^{2/3}\tilde{V}^{1/3} + T^{4/5}) + h(N)T$, where $\tilde{O}(T^{4/5})$ is the learning error for transition kernels and variation budget. It is proved in Tripathi & Modiano (2024) that $h(N) = 0$ for $N = 2$. Other prior works Gast et al. (2023); Verloop (2016); Weber & Weiss (1990); Gast et al. (2021) showed that $h(N) \to 0$ as $N \to \infty$ if the RMAB is indexable and has a global attractor point. This suggests that our policy can achieve sub-linear regret $\tilde{O}(T^{2/3}\tilde{V}^{1/3} + T^{4/5})$ for large system size and sub-linear $\tilde{V}$.*

| Applications | $(N, M)$ | Our Policy | UCWhittle | UCWhittle+Win | Random | WIQL |
|---|---|---|---|---|---|---|
| 1-D Bandit | (10, 1) | $\mathbf{957}_{\pm 155}$ | $6528_{\pm 1996}$ | $6377_{\pm 1452}$ | $11916_{\pm 2154}$ | $12060_{\pm 2226}$ |
| | (10,4) | $\mathbf{2119}_{\pm 237}$ | $27620_{\pm 5850}$ | $21628_{\pm 3054}$ | $28349_{\pm 4668}$ | $28068_{\pm 4601}$ |
| | (20,4) | $\mathbf{2065}_{\pm 368}$ | $28985_{\pm 7417}$ | $24405_{\pm 7286}$ | $39358_{\pm 6145}$ | $40314\text{-}_{\pm 6491}$ |
| Scheduling (Synthetic) | (10,1) | $\mathbf{503}_{\pm 31}$ | $981_{\pm 55}$ | $787_{\pm 42}$ | $3239_{\pm 115}$ | $3408_{\pm 98}$ |
| | (10, 4) | $\mathbf{945}_{\pm 94}$ | $1183_{\pm 152}$ | $1095_{\pm 118}$ | $2598_{\pm 49}$ | $2216_{\pm 64}$ |
| | (20,4) | $\mathbf{1276}_{\pm 90}$ | $2097_{\pm 189}$ | $1808_{\pm 138}$ | $7094_{\pm 189}$ | $6397_{\pm 279}$ |
| Scheduling (Real) | (6,1) | $\mathbf{2003}_{\pm 32}$ | $4821_{\pm 176}$ | $4772_{\pm 99}$ | $10539_{\pm 80}$ | $5171_{\pm 163}$ |
| | (6,3) | $\mathbf{1964}_{\pm 35}$ | $3544_{\pm 73}$ | $3504_{\pm 73}$ | $19587_{\pm 38}$ | $4163_{\pm 72}$ |

Table 1: $\mathrm{Reg}(T)$ for different values of $N$ and $M$.

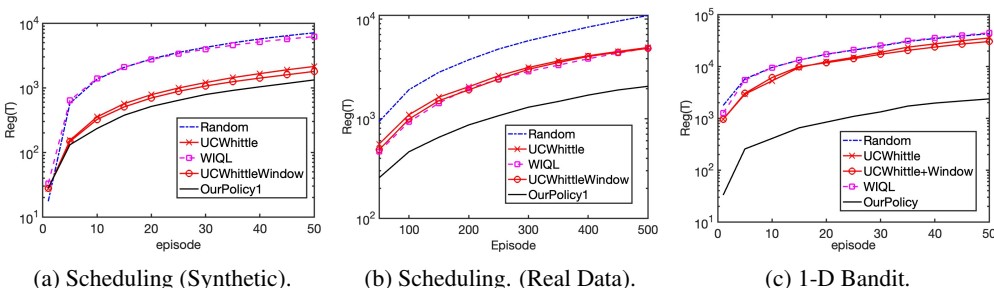

(a) Scheduling (Synthetic).    (b) Scheduling. (Real Data).    (c) 1-D Bandit.

Figure 1: $\mathrm{Reg}(\mathrm{T})$ Vs. number of episodes in Scheduling and 1-D Bandit.

# 6 SIMULATION RESULTS

In this section, we demonstrate the performance of our proposed policy by evaluating it under two applications (wireless scheduling and one-dimensional bandit) modeled as RMAB. For wireless scheduling, we provide performance evaluations using both synthetic and real-world datasets. In each application, we consider that there are $N$ arms and a policy can activate $M$ of them in each time slot $h \in [H]$ of every episode $t \in [T]$. We evaluate our policy against the UCWhittle policy Wang et al. (2023), UCWhittle + Window policy, where we incorporate sliding window to UCWhittle and the window size is taken randomly, the WIQL policy Biswas et al. (2021), and a randomized policy Kadota et al. (2018). The results are averaged over 50 independent runs. More details of our experimental setup are discussed in Appendix A.5.

Simulation results are shown in Table 1 and Figure 1. Our algorithm achieves the best regret in all cases. In contrast, UCWhittle is designed for static settings, uses all historical data and naïvely attempts to learn the entire transition matrix. While UCWhittle+Window employs sliding windows, the window size is chosen randomly and it does not utilize extra optimism for non-stationarity. WIQL, a Q-learning approach, requires extensive data samples to converge and uses all historical data. Specifically, our performance gain can be attributed to two main factors: (i) our intelligent update of the window size for predicting transition kernels, and (ii) our algorithm's exploitation of sparsity knowledge.

# 7 CONCLUSIONS AND LIMITATIONS

This paper introduced an online/reinforcement learning algorithm for estimating the Whittle index for restless bandit problems with unknown and non-stationary transition kernels using sliding window and upper confidence bound approaches. To our knowledge, this is the first work to provide an upper bound of the dynamic regret of an online Whittle index-based algorithm for RMABs with unknown and non-stationary transition kernels. Our proposed algorithm is evaluated on two different restless bandit problems against four baselines and provides significant performance gains. We also provide novel regret analysis. An interesting direction of future work involves proving lower bounds for regret. Other future directions include extending this work to infinite or continuous state spaces, and designing algorithms that achieve sub-linear dynamic regret even for large $\tilde{V}_n$ (rapidly varying kernels).

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

## A APPENDIX

### A.1 PROOF OF LEMMA 1

The L1-deviation of the true distribution and the empirical distribution of $m$ events is bounded by Weissman et al. (2003):

$$\Pr(|\hat{p} - p|_1 \geq \beta) \leq (2^m - 2)\exp^{(-\frac{k\beta^2}{2})},\tag{20}$$

where $k$ is the number of samples.

We denote $\mathbf{1}(s', s, a, n, t)$ as an indicator variable that represents the event of state $s$, action $a$, and next state $s'$ for arm $n$ at one time slot of episode $t$. Similarly, $\mathbf{1}(s', s, a, n, t, w)$ is an indicator variable that represents the event of state $s$, action $a$, and next state $s'$ for arm $n$ at one time slot in any one of the episodes $t - w + 1, t - (w-1) + 1, \ldots, t - 1$.

By using equation 20 with

$$\beta = \sqrt{\frac{2|\mathcal{S}|\log(2|\mathcal{S}||\mathcal{A}|NT/\eta)}{C_{t,w}^{(n)}(s,a)}}$$

and

$$k = C_{t,w}^{(n)}(s,a),$$

we get

$$\Pr\left(\|\hat{P}_{n,t,w}(\cdot|s,a) - \mathbb{E}[\mathbf{1}(\cdot, s, a, n, t, w)]\|_1 \geq \sqrt{\frac{2|\mathcal{S}|\log(2|\mathcal{S}||\mathcal{A}|NT/\eta)}{C_{t,w}^{(n)}(s,a)}}\right)$$
$$\leq \frac{\eta}{N|\mathcal{S}||\mathcal{A}|T}.\tag{21}$$

With probability one, we have

$$\|P_{n,t}(\cdot|s,a) - \mathbb{E}[\mathbf{1}(\cdot, s, a, n, t, w)]\|_1$$
$$\leq \|P_{n,t}(\cdot|s,a) - \max_{t' \in \{t-w+1, t-(w-1)+1, \ldots, t\}} \mathbb{E}[\mathbf{1}(\cdot, s, a, n, t', 1)]\|_1$$
$$= \|P_{n,t}(\cdot|s,a) - \max_{t' \in \{t-w+1, t-(w-1)+1, \ldots, t\}} P_{n,t'}(\cdot|s,a)]\|_1 \leq w_n V_n/T.\tag{22}$$

Now, by combining equation 21 and equation 22, we have

$$\Pr(P_{n,t} \in B_{n,t}, \forall n, \forall t) \geq 1 - \sum_{t=1}^{T} \sum_{n=1}^{N} \sum_{(s,a) \in \mathcal{S} \times A} \frac{\eta}{N|\mathcal{S}||\mathcal{A}|T}$$
$$= 1 - \eta.\tag{23}$$

This concludes the proof of Lemma 1.

### A.2 PROOF OF THEOREM 1

We first decompose Term1. We solve another bandit problem to select $V_n$ from a set of possible drift values based on the history by using EXP3 algorithm Auer et al. (2002). In this case, we can decompose the regret associated with Term1 as follows:

$$\text{Term1} = \sum_{t=1}^{T} \sum_{n=1}^{N} U(\pi_{n,t}^*, P_{n,t}, \lambda) - U(\pi_{n,t}(\hat{V}_n(t)), P_{n,t}, \lambda)$$
$$= \sum_{t=1}^{T} \sum_{n=1}^{N} \left( U(\pi_{n,t}^*, P_{n,t}, \lambda) - U(\pi_{n,t}(V_n), P_{n,t}, \lambda) \right)$$
$$+ \sum_{t=1}^{T} \sum_{n=1}^{N} \left( U(\pi_{n,t}(V_n), P_{n,t}, \lambda) - U(\pi_{n,t}(\hat{V}_n(t)), P_{n,t}, \lambda) \right)\tag{24}$$

where we denote $V_n$ is optimal, $\hat{V}_n(t)$ is estimated, and $\pi_n(\mathcal{V})$ denotes our policy when $\mathcal{V}$ is used.

The regret bound for the term

$$\sum_{t=1}^{T}\sum_{n=1}^{N}\left(U(\pi_{n,t}(V_n),P_{n,t},\lambda)-U(\pi_{n,t}(\hat{V}_n(t)),P_{n,t},\lambda)\right)$$

represents the loss from needing to learn $V_n$ instead of knowing it apriori and can be found by using Auer et al. (2002). In particular, assuming the finite discretization of possible drifts in Section 4.3, we can have

$$\sum_{t=1}^{T}\sum_{n=1}^{N}\left(U(\pi_{n,t}(V_n),P_{n,t},\lambda)-U(\pi_{n,t}(\hat{V}_n(t)),P_{n,t},\lambda)\right)\leq\sum_{n=1}^{N}O\left(\sqrt{J_n\log(J_n)T}\right),\quad(25)$$

where $J_n$ is the number of elements in the set of possible values of drift for arm $n$.

Next, we show the upper bound of $\sum_{t=1}^{T}\sum_{n=1}^{N}\left(U(\pi_{n,t}^*,P_{n,t},\lambda)-U(\pi_{n,t}(V_n),P_{n,t},\lambda)\right)$. For the ease of notation, we use $\pi_{n,t}(V_n)$ as $\pi_{n,t}$. When the confidence bound holds, we have

$$\sum_{n=1}^{N}U(\pi_{n,t}^*,P_{n,t},\lambda)-U(\pi_{n,t},P_{n,t},\lambda)\leq\sum_{n=1}^{N}U(\pi_{n,t},\tilde{P}_{n,t},\lambda)-U(\pi_{n,t},P_{n,t},\lambda)$$

$$\overset{a}{=}\sum_{n=1}^{N}\mathbb{E}_{P_{n,t},\pi_{n,t}}\left[\sum_{h=1}^{H}\sum_{s'\in\mathcal{S}}\gamma^{h-1}(\tilde{P}_{n,t}(s'|s_{n,t,h},a_{n,t,h})-P_{n,t}(s'|s_{n,t,h},a_{n,t,h}))V_n(s';\pi_{n,t},\tilde{P}_{n,t})\right],$$

$$\leq\sum_{n=1}^{N}\mathbb{E}_{P_{n,t},\pi_{n,t}}\left[\sum_{(s,a)\in\mathcal{S}}\alpha_t^{(n)}(s,a)\sum_{s'\in\mathcal{S}}\left|\tilde{P}_{n,t}(s'|s,a)-P_{n,t}(s'|s,a)\right|\right]V_{max},$$

$$\leq\sum_{n=1}^{N}\mathbb{E}_{P_{n,t},\pi_{n,t}}\left[\sum_{(s,a)\in\mathcal{S}\times\mathcal{A}}\alpha_t^{(n)}(s,a)d_t^{(n)}(s,a)\right]V_{max}\qquad(26)$$

where (a) is obtained by using (Wang et al., 2023, Theorem 6.4), the simplified notation $V_n$ is used instead of $V_{n,\lambda,t}$, $V_{max}=\max_{n\in[N],s\in\mathcal{S}}V_n(s';\pi_{n,t},\tilde{P}_{n,t})$ and $\alpha_t^{(n)}(s,a)$ is a random variable denoting the number of visits of $(s,a)\in\mathcal{S}\times\mathcal{A}$ at episode $t\in[T]$.

By substituting the value of $d_t^{(n)}(s,a)$, we have

$$\sum_{t=1}^{T}\sum_{n=1}^{N}\mathbb{E}_{P_{n,t},\pi_{n,t}}\left[\sum_{(s,a)\in\mathcal{S}\times\mathcal{A}}\alpha_t^{(n)}(s,a)d_t^{(n)}(s,a)\right]$$

$$\leq\sum_{t=1}^{T}\left(\sum_{n=1}^{N}\sqrt{2|\mathcal{S}|\log(2|\mathcal{S}||\mathcal{A}|NT/\eta)}\mathbb{E}_{P_{n,t},\pi_{n,t}}\left[\sum_{(s,a)\in\mathcal{Z}_2}\frac{\alpha_t^{(n)}(s,a)}{\sqrt{C_{t,w_n}^{(n)}(s,a)}}\right]+w_nV_n/TH\right)$$

$$=\sum_{t=1}^{T}\left(\sum_{n=1}^{N}\sqrt{2|\mathcal{S}|\log(2|\mathcal{S}||\mathcal{A}|NT/\eta)}\sum_{(s,a)\in\mathcal{S}\times\mathcal{A}}g_{t,n}(s,a,w_n)+w_nV_nH/T\right)$$

$$\leq\sum_{t=1}^{T}\left(\sum_{n=1}^{N}\sqrt{2|\mathcal{S}|\log(2|\mathcal{S}||\mathcal{A}|NT/\eta)}2|\mathcal{S}|G_{t,n}(w_n)+w_nV_nH/T\right)$$

$$=\sum_{t=1}^{T}O\left(\sum_{n=1}^{N}2|\mathcal{S}|G_{t,n}(w_n)+w_nV_nH/T\right),\qquad(27)$$

where $G_{t,n}(w)=\max_{(s,a)\in\mathcal{S}\times\mathcal{A}}g_{t,n}(s,a,w)$ and

$$g_{t,n}(s,a,w)=\mathbb{E}_{P_{n,t},\pi_{n,t}}\left[\sum_{(s,a)\in\mathcal{Z}_2}\frac{\alpha_t^{(n)}(s,a)}{\sqrt{C_{t,w}^{(n)}(s,a)}}\right]$$

is a non-increasing function of the window size $w$. This is because $C_{t,w}^{(n)}(s,a)$ is a non-decreasing function of $w$.

Now, in the above analysis $V_n$ is the optimal choice of drift values. Specifically, the optimal choice of $V_n$ from the set of drift values satisfies: $\max_{(s,a)\in\mathcal{S}\times\mathcal{A}}\sum_{s'\in\mathcal{S}}\left|P_{n,t}(s'|s,a)-P_{n,t-1}(s'|s,a)\right|\leq$

$V_n/T\leq\max_{(s,a)\in\mathcal{S}\times\mathcal{A}}\sum_{s'\in\mathcal{S}}\left|P_{n,t}(s'|s,a)-P_{n,t-1}(s'|s,a)\right|+2/J_nT$. Consequently, the optimal upper bound of total variation budget $V_{n,T}$ satisfies

$$\tilde{V}_n\leq V_n\leq\tilde{V}_n+2T/J_n.$$

Then, the upper bound becomes

$$\sum_{t=1}^{T}O\left(\sum_{n=1}^{N}2|\mathcal{S}|G_{t,n}(w_n)\right)+\sum_{n=1}^{N}O\left(w_n(\tilde{V}_n+2T/J_n)H\right).$$

### A.3 PROOF OF THEOREM 2

Lets denote the probability to visit every $(s,a)\in\mathcal{S}\times\mathcal{A}$ at least once in an episode for all arms $n\in[N]$ by $P_{\min}$. According to the condition in Theorem 2, $P_{\min}>0$.

Now, to prove Theorem 2, we bound

$$g_{t,n}(s,a,w)=\mathbb{E}_{P_{n,t},\pi_{n,t}}\left[\frac{\alpha_t^{(n)}(s,a)}{\sqrt{C_{t,w}^{(n)}(s,a)}}\right]\leq H\mathbb{E}_{P_{n,t},\pi_{n,t}}\left[\frac{1}{\sqrt{C_{t,w}^{(n)}(s,a)}}\right], \tag{28}$$

where the number of visit $\alpha_t^{(n)}(s,a)$ in one episode is upper bounded by the time horizon $H$.

Let $\mathbb{E}[C_{t,w}^{(n)}(s,a)]=\mu$. Then, $\mu\geq 1$ because by definition, $C_{t,w}^{(n)}(s,a):=\max\left\{\sum_{s'\in\mathcal{S}}C_{t,w}^{(n)}(s',a,s),1\right\}$. Moreover, $\mu\geq w_nP_{\min}$.

Now, we have

$$\mathbb{E}\left[\frac{1}{\sqrt{C_{t,w}^{(n)}(s,a)}}\right]=\mathbb{E}\left[\frac{1}{\sqrt{C_{t,w}^{(n)}(s,a)}}\bigg|C_{t,w}^{(n)}(s,a)<\frac{\mu}{2}\right]P\left(C_{t,w}^{(n)}(s,a)<\frac{\mu}{2}\right)$$

$$+\mathbb{E}\left[\frac{1}{\sqrt{C_{t,w}^{(n)}(s,a)}}\bigg|C_{t,w}^{(n)}(s,a)\geq\frac{\mu}{2}\right]P\left(C_{t,w}^{(n)}(s,a)\geq\frac{\mu}{2}\right) \tag{29}$$

If $C_{t,w}^{(n)}(s,a)\geq\mu/2$, then $1/\sqrt{C_{t,w}^{(n)}(s,a)}\leq 1/\sqrt{\mu/2}=\sqrt{2/\mu}$. This part of the expectation is therefore bounded by $\sqrt{2/\mu}\cdot P(C_{t,w}^{(n)}(s,a)\geq\mu/2)\leq\sqrt{2/\mu}\leq\sqrt{\frac{2}{w_nP_{min}}}$.

If $C_{t,w}^{(n)}(s,a)<\mu/2$, there exists a constant $\eta>0$ such that we have $P(C_{t,w}^{(n)}(s,a)<(1-1/2)\mu)\leq O(e^{-\mu/4\eta})\leq O(e^{-w_nP_{\min}/4\eta})$ by using the Chernoff bound for Markov Chains Chung et al. (2012).

Thus, the expectation becomes

$$\mathbb{E}\left[\frac{1}{\sqrt{C_{t,w}^{(n)}(s,a)}}\right]\leq\frac{\sqrt{2}}{\sqrt{w_nP_{\min}}}+O(e^{-w_nP_{\min}/4\eta}). \tag{30}$$

We can have a constant $\eta_1>0$ independent of $w_n$ and $P_{\min}$ such that

$$\frac{\sqrt{2}}{\sqrt{w_nP_{\min}}}+e^{-w_nP_{\min}/4\eta}\leq\frac{\eta_1}{\sqrt{w_nP_{\min}}}=O(1/\sqrt{w_nP_{\min}}). \tag{31}$$

Therefore, we have

$$g_{t,n}(s,a,w) = \mathbb{E}_{P_{n,t},\pi_{n,t}} \left[ \frac{\alpha_t^{(n)}(s,a)}{\sqrt{C_{t,w}^{(n)}(s,a)}} \right] \leq O\left( \frac{H}{\sqrt{w_n P_{\min}}} \right) \tag{32}$$

Next, we have

$$\sum_{t=1}^{T} \left( \frac{H}{\sqrt{w_n P_{\min}}} + \frac{w_n V_n H}{T} \right) = H\left( \frac{T}{\sqrt{w_n P_{\min}}} + V_n w_n \right). \tag{33}$$

Then, by substituting $w_n = (T/V_n)^{2/3}$, we get

$$H\left( \frac{T}{\sqrt{w_n P_{min}}} + w_n V_n \right) = HT^{2/3} V_n^{1/3} P_{\min}^{-1/2} + HT^{2/3} V_n^{1/3} = \tilde{O}\left( T^{2/3} V_n^{1/3} \right), \tag{34}$$

where $H$ and $P_{\min}$ are absorbed in big-O-notation because $H$ is constant number time-slots in every episode, $P_{\min}$ depends on the number of time-slots $H$ and the initial state in any episode.

By substituting $V_n = \tilde{V}_n + 2T/J_n$ in the above, we obtain Theorem 2.

### A.4 PROOF OF THEOREM 3

We first decompose Term2. By adding and subtracting

$$\sum_{n=1}^{N} U(\pi_{n,t}, \tilde{P}_{n,t}, \lambda) \text{ and } R_t\left( \pi_t, (\tilde{P}_{n,t})_{n=1}^{N} \right),$$

we can express Term2 as follows:

$$\sum_{t=1}^{T} \left( \sum_{n=1}^{N} U\left( \pi_{n,t}, P_{n,t}, \lambda \right) - R_t\left( \pi_t, (P_{n,t})_{n=1}^{N} \right) \right)$$

$$= h(N)T + \sum_{t=1}^{T} \left( \sum_{n=1}^{N} U(\pi_{n,t}, P_{n,t}, \lambda) - \sum_{n=1}^{N} U(\pi_{n,t}, \tilde{P}_{n,t}, \lambda) \right.$$

$$\left. + R_t\left( \pi_t, (\tilde{P}_{n,t})_{n=1}^{N} \right) - R_t\left( \pi_t, (P_{n,t})_{n=1}^{N} \right). \tag{35}$$

When the confidence bound holds,

$$U\left( \pi_{n,t}, P_{n,t}, \lambda \right) - U\left( \pi_{n,t}, \tilde{P}_{n,t}, \lambda \right) \leq 0. \tag{36}$$

This is because $\pi_{n,t}$ is the optimal solution of the Lagrangian problem and $\tilde{P}_{n,t}$ achieves the highest Lagrangian objective value.

Similar to equation 26(a), by using (Wang et al., 2023, Theorem 6.4), we can have

$$R_t\left( \pi_t, (\tilde{P}_{n,t})_{n=1}^{N} \right) - R_t\left( \pi_t, (P_{n,t})_{n=1}^{N} \right)$$

$$= \mathbb{E}_{P_t,\pi_t} \left[ \sum_{n=1}^{N} \sum_{h=1}^{H} \sum_{s' \in \mathcal{S}} \gamma^{h-1} (\tilde{P}_{n,t}(s'|s_{n,t,h}, a_{n,t,h}) - P_{n,t}(s'|s_{n,t,h}, a_{n,t,h})) V_n(s'; \pi_t, \tilde{P}_{n,t}) \right]$$

$$\leq \mathbb{E}_{P_t,\pi_t} \left[ \sum_{n=1}^{N} \sum_{(s,a)\mathcal{S}\times\mathcal{A}} \alpha_t^{(n)}(s,a) d_t^{(n)}(s,a) \right] V_{max} \tag{37}$$

which is similar to the last inequality of equation 26. Hence, similar to Theorem 1 and Theorem 2, we can have

$$R_t\left( \pi_t, (\tilde{P}_{n,t})_{n=1}^{N} \right) - R_t\left( \pi_t, (P_{n,t})_{n=1}^{N} \right) \leq \tilde{O}(T^{2/3}(\tilde{V}_n + 2T/J)^{1/3}) + \tilde{O}(\sqrt{TJ_n}) \tag{38}$$

Therefore, we have

$$\text{Reg(T)} \leq \text{Term1} + \text{Term2}$$

$$\leq \sum_{n=1}^{N} \tilde{O}(2T^{2/3}(\tilde{V}_n + 2T/J)^{1/3}) + \sum_{n=1}^{N} \tilde{O}(2\sqrt{TJ_n}) + h(N)T.$$

Next, we can substitute $J_n = O(T^{3/5})$ and obtain

$$\tilde{O}(2T^{2/3}(\tilde{V}_n + 2T/J)^{1/3}) + \tilde{O}(2\sqrt{TJ_n})$$
$$\leq \tilde{O}(2T^{2/3}\tilde{V}_n^{1/3}) + \tilde{O}(2^{4/3}T^{4/5}) = \tilde{O}(T^{2/3}\tilde{V}_n^{1/3} + T^{4/5}) \tag{39}$$

Next, by using $\tilde{V} = \max_n \tilde{V}_n$, we have $\sum_{n=1}^{N} \tilde{O}(T^{2/3}\tilde{V}_n^{1/3} + T^{4/5}) = \tilde{O}(T^{2/3}\tilde{V}^{1/3} + T^{4/5})$. This concludes the proof.

## A.5 EXPERIMENTAL SETUP

Firstly, we discuss how we compute the regret in our numerical studies. Because it is not possible to obtain the optimal RMAB policy even if everything is known exactly, we use the best Whittle index policy to demonstrate the regret in Table 1 and Fig. 1. Given a policy, we evaluate the regret of the policy at each episode $t$ by subtracting the total discounted sum reward of all arms obtained by the policy from the total discounted sum reward of all arms received by a Whittle index policy with access to the true transition probabilities. In our simulation, we have used MATLAB. In simulation results of Table 1 and Fig. 1, a discount factor of $\gamma = 0.99$ was considered. Time-slots $H = 50$ and $T = 50$ episodes were considered for wireless scheduling (synthetic) and 1-D bandit. Time-slots $H = 500$ and $T = 500$ episodes were considered for wireless scheduling (Real). Moreover, we considered $V_n = V$ for all $n$ in the wireless scheduling (synthetic) and 1-D bandit problems. But, in wireless scheduling (Real), $V_n$ can vary across $n$ and depends on the dataset. In Figure 1, we used $N = 20, M = 4$ for wireless scheduling (synthetic) and 1-D bandit. For wireless scheduling (Real), we consider $N = 6$ and $M = 1$.

Now, we discuss how we model One Dimensional Bandits and Wireless Scheduling.

**One Dimensional Bandits:** We consider a modified version of the one dimensional RMAB problem studied in Killian et al. (2021); Nakhleh et al. (2022). Each arm $n$ is a Markov process with $K$ states, numbered as $0, 1, \ldots, K - 1$. For our simulations, we set $K = 10$. The reward of an arm increases linearly with the current state, i.e. $r(s, a) = s$. If the arm is activated, then it can evolve from state $s$ to $\min\{s + 1, K - 1\}$ with probability $q_n(t)$ or remain in the same state $s$ with probability $1 - q_n(t)$. If the arm is not activated, then it evolves from state $s$ to $\max\{s - 1, 0\}$ with probability $p_n(t)$ or remain in the same state $s$ with probability $1 - p_n(t)$. One-dimensional MDPs of this form are often used in health monitoring and machine monitoring applications Matsena Zingoni et al. (2021); Parisi et al. (2024). In our simulation, we consider - (i) $V_n = 35$, (ii) $p_n(t)$ changes to $\min\{p_n(t-1) + \frac{V_n}{4T}, 1\}$ with probability 0.5, or it changes to $\max\{p_n(t-1) - \frac{V_n}{4T}, 0\}$ with probability 0.5 and (iii) $q_n(t)$ changes to $\min\{q_n(t-1) + \frac{V_n}{4T}, 1\}$ with probability 0.5, or it changes to $\max\{q_n(t-1) - \frac{V_n}{4T}, 0\}$ with probability 0.5.

**Wireless Scheduling Using Synthetic Data:** We consider a wireless scheduling problem, where $M$ out of $N$ sources can send their observation to a receiver side over an unreliable channel at every time slot $h \in [H]$ of episode $t \in [T]$. Due to channel unreliability, the observation may not be delivered. The goal of the receiver is to estimate the current signal values of all $N$ sources based the information delivered from the sources. The reward for accurate timely estimation can be modeled as the mutual information between the estimated signal and the actual signal. Sun & Cyr (2019) showed that the mutual information can be determined by using a decreasing function $-(\log_2(1 - \sigma_n^{2s_{n,h,t}}))/2$ of Age of Information (AoI) for zero-mean i.i.d. Gaussian random variables with variance $\sigma_n^2$, where AoI $s_{n,h,t}$ of source $n$ is the time difference between current time $h$ and the generation time of the most recently delivered signal. The AoI value of a source $n$ increases by 1 if the source $n$ is not scheduled. If the source $n$ is scheduled, the AoI value drops to 1 with probability $q_n(t)$ (successful delivery) or increases by 1 with probability $1 - q_n(t)$ (unsuccessful delivery). The parameter $q_n(t)$ measures the reliability of channel $n$ at time $t$. In our experiment, we assume that $q_n(t)$ is unknown

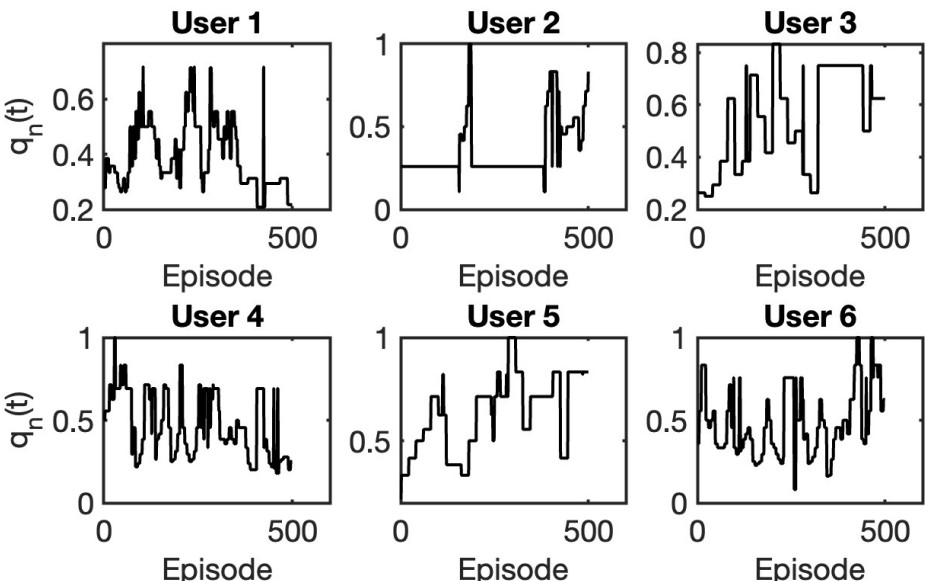

Figure 2: Success Probability $q_n(t)$ in 500 episodes

and non-stationary for half of the sources, whereas it is is unknown but stationary for the remaining half. For non-stationary arms, the variance of signal values $\sigma_n^2 = 0.9$ is used and the probability of successful transmission $q_n(t)$ changes to $\min\{q_n(t-1) + \frac{V_n}{2T}, 1\}$ with probability 0.6, or it changes to $\max\{q_n(t-1) - \frac{V_n}{2T}, 0\}$ with probability 0.4; the initial value of $q_n(t) = 0.1$ is used. For the other half, $q_n(t) = 1$ is unknown but stationary and $\sigma_n^2 = 0.5$.

**Wireless Scheduling Using Real-world Dataset:**

We have also incorporated a recent dataset from Reddy et al. (2025) for the wireless scheduling problem. The dataset contains traces of measured signal strength for six users across indoor and outdoor settings, leading to non-stationary behavior. The signal strength time-series values allow us to calculate packet transmission success probabilities, which we then utilize to set up our wireless scheduling problem. In Figure 2, we plot these success probability values, which clearly demonstrate the highly time-varying nature of the dataset due to user mobility. We also plot variation $\epsilon_n(t) = |q_n(t) - q_n(t-1)|$ in success probability in Figure 3. In this simulation, we consider six users with AoI function $-a\log 2(1 - 0.9^{s_{n,h,t}})$, where we set $a = 0.4$ for three users, $a = 0.5$ for one user, $a = 0.9$ for the other two users.

A.6 COMPUTATIONAL COMPLEXITY

At the beginning of every episode $t$, we must compute the predicted transition kernels for all arms by solving the optimization problem equation 12. The solution can be obtained either via a closed-form expression or by employing the Extended Value Iteration (EVI) algorithm Auer et al. (2008). Crucially, this computation is performed only once per episode, not at every time step. When a closed-form solution is unavailable, each iteration of the EVI algorithm requires $O(|\mathcal{S}|^2|\mathcal{A}|)$ time per state $s \in \mathcal{S}$. Subsequently, at every time step within the episode, the Whittle index for all states is computed. This can be achieved either through a straightforward closed-form equation or by iteratively solving equation 7 using the bisection method. With a specified tolerance $\mathrm{tol}$, an upper bound $u$, and a lower bound $l$, the bisection method requires at least $O(\log_2((u-l)/\mathrm{tol}))$ steps per state. We now proceed to a detailed analysis of the computational complexity associated with solving equation 12.

In many problems, for example in wireless scheduling and one dimensional bandit problem, we can get closed form solution of equation 12, making the bilinear optimization very efficient. Here are the closed form solution:

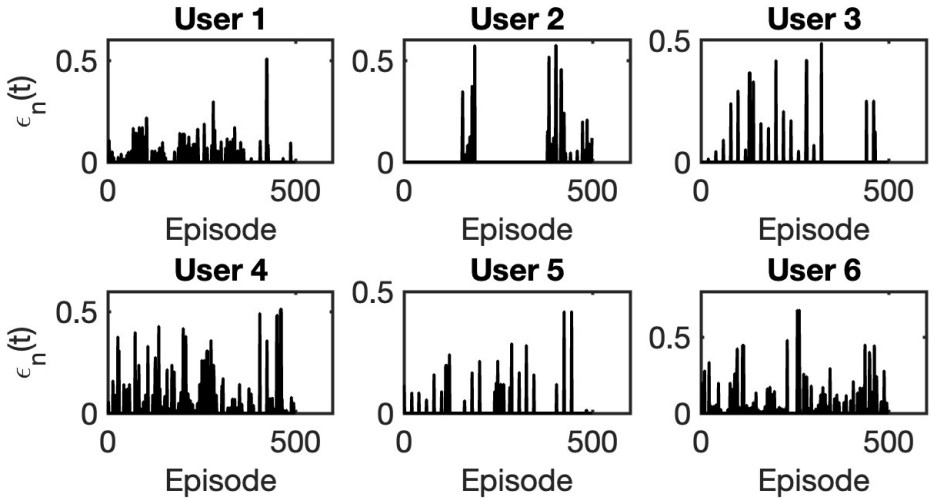

Figure 3: Variation $\epsilon_n(t) = |q_n(t) - q_n(t-1)|$ in 500 Episodes

• Wireless Scheduling:

$$P_{n,t}(1|s,1) = \min\left\{\hat{P}_{n,t,w_i}(1|s,1) + d_t^{(n)}(s,1)/2, 1\right\}, \tag{40}$$

$$P_{n,t}(s+1|s,1) = 1 - P_{n,t}(1|s,1), \tag{41}$$
$$P_{n,t}(s+1|s,0) = 1. \tag{42}$$

• One Dimensional Bandit:

$$P_{n,t}(s+1|s,1) = \min\left\{\hat{P}_{n,t,w_i}(s+1|s,1) + d_t^{(n)}(s,1)/2, 1\right\}, \tag{43}$$

$$P_{n,t}(s|s,1) = 1 - P_{n,t}(s+1|s,1), \tag{44}$$

$$P_{n,t}(s|s,0) = \min\left\{\hat{P}_{n,t,w_i}(s|s,0) + d_t^{(n)}(s,0)/2, 1\right\}, \tag{45}$$

$$P_{n,t}(s-1|s,0) = 1 - P_{n,t}(s-1|s,0), \tag{46}$$

In settings where we do not have closed form solutions, we can use extended value iteration algorithm Auer et al. (2008). In the extended value iteration algorithm, in every iteration, we visit every state $s \in \mathcal{S}$ to update the value function. The complexity to update the value function for each $s \in \mathcal{S}$ in the extended value iteration algorithm is $O(|\mathcal{S}|^2|\mathcal{A}|)$, whereas the complexity of value iteration algorithm is $O(|\mathcal{S}||\mathcal{A}|)$, where $\mathcal{S}$ and $|\mathcal{A}|$ are total number of states and actions, respectively. The extra computation, we need is to solve the linear problem

$$\max_{P_{n,t} \in B_t^{(n)}} \sum_{s' \in \mathcal{S}} P_{n,t}(s'|s,a) V_{n,t,\lambda}(s'),$$

which takes $O(|\mathcal{S}|)$ time.

**Wall Clock Time:** For wireless scheduling problem, we have count the wall clock time. One iteration in extended value iteration algorithm takes 0.037 sec and one iteration using closed form solution takes 0.006 sec. These are implemented using MATLAB in MacBook Pro, 2022 with Apple M2 chip and 8 GB memory.

Next, at every time step $h$ of each episode $t$, we need to sort $N$ arms using Whittle index and select $M$ arms with highest Whittle indices. Sorting can take $O(N\log N)$ time.

**How Sparsity Helps:** The sparsity helps us find computationally efficient solutions to equation 12. The complexity to update the value function for each $s \in \mathcal{S}$ reduces to $O((|\mathcal{S} - \mathcal{S}(s,a)|)^2|\mathcal{A}|)$, if we know the sparsity information $\mathcal{S}(s,a)$.

---

**Algorithm 2:** BoB algorithm for choosing $V$

---

**input :** $\beta \in (0, 1]$, $J = O(T^{3/5})$ (we omit subscript $n$ from $J_n$ and $V_n$)

1   Initialize $V(i) = V_{max} - (i - 1)V_{max}/J$

2   Initialize $w_i = 1$ and $\hat{X}_i = 0$

3   **for** *every episode* $t = 1, 2, \ldots, T$ **do**

4     Set $p_i(t) = (1 - \beta)\frac{w_i}{\sum_{i=1}^{J} w_i} + \frac{\beta}{J}$

5     Select $i_t \in \{1, \ldots, J\}$ randomly according to probability $p_1(t), \ldots, p_J(t)$, respectively

6     Select $V(i_t)$ at episode $t$ and Observe Reward $R_{n,t}$

7     Normalize reward: $X_t = R_{n,t}/r_{max}H$, where $r_{max} = \max_{s,a,n} r_n(s,a)$

8     Update $\hat{X}_{i_t} \leftarrow X_t/p_{i_t}(t)$

9     Update $w_i \leftarrow w_i \exp(\beta\hat{X}_i/J)$

---

| $(N, M)$ | $V_n$ known | $J_n = 40$ | $J_n = 20$ | $J_n = 10$ |
|---|---|---|---|---|
| $(6, 1)$ | 1878 | 2003 | 2024 | 2101 |
| $(6, 3)$ | 1897 | 1964 | 1956 | 1934 |

Table 2: $\text{Reg}(T)$ for known $V_n$ and different $J_n$ with unknown $V_n$ for Scheduling Problem (Real Dataset).

## A.7   IMPACT OF THE BOB ALGORITHM

Now, we discuss the impact of the BoB algorithm provided in Algorithm 2 on the regret, specifically the impact of the parameter $J_n$. Theorem 2 directly quantifies the fundamental trade-off introduced by the quantization level in the BoB approach: increasing $J_n$ improves the accuracy of tracking the variation budget $V_n$ but concurrently slows down the BoB learning. The effectiveness of the approach is empirically evaluated in the new Table 2, which shows that performance is robust and not highly sensitive to the exact value of $J_n$, provided a sufficiently large level is chosen.

