# OpenReview forum: "Online Learning of Whittle Indices for Restless Bandits with Non-Stationary Transition Kernels"
_ICLR.cc/2026/Conference — Submitted to ICLR 2026_

### Official Review · Reviewer_KPF4 · 2025-10-31

**Soundness:** 3
**Presentation:** 3
**Contribution:** 2
**Rating:** 4
**Confidence:** 3

**Summary:**

This work studies a non-stationary variant of the restless multi-armed bandit (RMAB) problem where (unknown) transition dynamics change over time. They adapt standard non-stationary sliding-window and bandit-over-bandit meta-algorithm approaches on the Whittle index-based policy for RMAB to get dynamic regret bounds scaling with the total variation in changes of the transition matrices.

**Strengths:**

* The paper provides the first dynamic regret bounds for the non-stationary RMAB setting.
* The presentation is clear and easy to follow.
* There are simulation results validating the theoretical results and showing superior regret of their algorithm over other arts.

**Weaknesses:**

* The work [1] also seems to establish dynamic regret bounds for non-stationary RMAB, albeit with larger scaling in terms of $T$ and variation budget. It would be good to discuss the differences in approaches and results.
* It is unclear to me if the dynamic regret bound shown in this work $O(T^{2/3} \tilde{V}^{1/3} + T^{4/5})$ is optimal. In particular, is the $T^{4/5}$ term an artifact of the analysis or in fact fundamental?
* Furthermore, it is unclear if the right dependence on other problem-dependent parameters such as $H,P_{\min}$ and the number of states has been captured optimally in the regret bounds as compared to the known stationary regret bounds for RMAB.
* The techniques used in this work (sliding-window and bandit-over-bandit) are fairly standard in works on non-stationary bandits and so the technical novelty of the submission is not very high I feel. Could the authors elaborate on what particular proof challenges or hurdles were overcome for this setting?

[1] Non-Stationary Restless Multi-Armed Bandits with Provable Guarantee. Hung et al., 2025. https://arxiv.org/pdf/2508.10804

**Questions:**

Please see weaknesses above.

---

> ### Author Response · Authors · 2025-11-21
> **Response to Weakness and Questions of Reviewer KPF4**
>
> We thank the reviewer for the constructive feedback. Below, we address the specific questions raised. We have updated the paper accordingly.
>
> **optimal regret bound:** The $T^{4/5}$ term in the regret bound specifically arises for the scenario where the variation budget is unknown: for known total variation budget, the term vanishes and we achieve $T^{2/3}$ regret bound, which is comparable to other MDP works with non-stationary transition kernels (Ortner et al. (2020)). This $T^{4/5}$ term arises directly from our particular choice for the set of possible variation budgets used in the Bandit over Bandit (BoB) approach. A different design for this set could yield a different term. To establish whether our bound is truly optimal, a further analysis of the regret lower bound for any general algorithm is required. Proving such lower bounds in non-stationary MDP settings is a complex challenge and an open area of exploration. This is a promising direction of future work for us, to support the theoretical claims in the current work. We have added discussion in Section 7 to reflect this.
>
> **Comparison to known stationary regret bounds for RMAB:** (Wang et. al., 2023) is the only paper that provides stationary regret bounds for RMAB. Compared to (Wang et. al., 2023), we have the same dependence in $H$ and the number of states, i.e., linear in $H$ and $|\mathcal S|$, as provided in our Theorem 1. $P_{min}$ is introduced in our paper with the assumption $P_{min}>0$ for handling non-stationary analysis.
>
> **Novelty and Challenges:** Yes, we agree that Sliding window and Bandit-over-Bandit techniques are used in MDPs under non-stationarity. Importantly RMABs are different from general MDPs because i) they suffer from the curse of dimensionality and finding optimal policies is hard even when all system parameters are stationary and known, ii) they have extra structure, i.e. there exist solution techniques like the Whittle Index which allow for near optimal policy design. The challenge for designing online learning algorithms for RMABs is to incorporate this computationally efficient class of policies into an adaptive process. The challenge for dynamic regret analysis is that the policies are only approximately optimal, even when everything is known, so characterizing and proving sublinear regret is tricky. We first resolved the question of designing confidence bounds. We found that we need to add an extra term related to drift and the window size to tackle the challenge. Our analysis in Lemma 1 then provides the probability when this confidence bound holds. Our proof technique is independent and different from prior works. Next, because of the online setting, the optimal policy changes over episodes. Further, due to the RMAB setting and the addition of the new term, the regret analysis becomes quite challenging compared to prior stationary work (Wang et. al. 2023). The main questions we resolve in Theore1 are: (i) how to analyze the impact of drift and window size on the dynamic regret? and (ii) how to obtain sub-linear regret? To answer the first question (i), in Theorem 1, we show that there is a trade-off between smaller window sizes and larger window size. Moreover, we define a new term $G_{t,n}(w_n)$ which characterizes loss for insufficient exploration. This term $G_{t,n}(w_n)$ has become a useful guide to designing new online learning policies. To answer the second question (ii), in Theorem 2, we design window sizes optimally. The design of the window size requires additional novel proof techniques. Specifically, we faced challenges to upper bound our newly introduced term $G_{t,n}(w_n)$ with respect to the window size $w_n$. To solve this challenge, we introduced a new assumption on the probability of visit to every state action pair with minimum probability $P_{min}>0$ and used Chernoff bound for Markov chains. After that, we have Theorem 3, where we worked on stronger definition of regret compared to (wang et. al. 2023) by using the main problem instead of using the Lagrangian relaxed problem. In working with stronger definition of regret, we separated a term $h(\cdot)$ that measures the gap between an optimal policy and a Whittle policy in a stationary setting, and is typically small and converging to zero for large classes of RMABs under global attractor and indexablity conditions.
>
> Please let us know if you have any further questions.

---

### Official Review · Reviewer_bw55 · 2025-11-01

**Soundness:** 3
**Presentation:** 3
**Contribution:** 3
**Rating:** 6
**Confidence:** 3

**Summary:**

The paper studies restless multi-armed bandits (RMABs) with unknown, time-varying transition kernels and proposes a sliding-window online Whittle policy. Each episode: (i) predict a per-arm variation budget $V_n$
 via a Bandit-over-Bandit (EXP3) scheme; (ii) set window size $w_n$; (iii) build a confidence set for transitions over the last $w_n$ episodes; (iv) pick an optimistic transition kernel $\tilde P_{n,t}$ within the confidence set that maximizes the arm’s value (thus upper-bounding the true value); (v) compute Whittle indices under $\tilde P_{n,t}$ and activate the top-$M$ arms; (vi) update the Lagrange multiplier $\lambda$ to the $M$-th largest index.

**Strengths:**

1. Timely problem \& clear algorithmic design:
The paper tackles online RMABs with non-stationary kernels—highly relevant in networking, recommendation, and health monitoring—and adapts Whittle policies via sliding windows + optimism + BoB for $V_n$.
The step-by-step procedure is explicit and implementable.

2. Non-stationarity modeling via variation budgets:
Using a per-arm total variation metric aligns with dynamic regret literature and naturally drives the windowing trade-off.

3. Confidence-set optimism specialized to RMABs:
The optimistic kernel selection within a high-dimensional ball, then computing Whittle indices on $\tilde P_{n,t}$, is principled and lets the value function upper-bound enable regret control.

4. S.4. Regret decomposition that exposes the right knobs:
The bound shows how (i) estimation error decreases with $w_n$, (ii) prediction/drift error grows with $\tilde V_n$ and the BoB granularity $J_n$, and (iii) the Lagrangian-relaxation gap contributes $h(N)T$—clarifying where hardness comes from and when sublinearity holds.

5. Sparsity awareness:
If the decision maker knows sets $S_0(s,a)$ of impossible transitions, the confidence region shrinks and the bound improves; yet the method still guarantees sublinear regret even without sparsity.

6. Empirical evidence across two RMABs.
Against strong baselines, the method consistently achieves lower regret, with a transparent evaluation protocol (oracle–Whittle comparator).

**Weaknesses:**

1. Additive linear term $h(N)T$ can dominate: The dynamic-regret bound contains an additive term $h(N)T$, where $h(N)$ measures the inherent performance gap between the Lagrangian (decoupled) problem and the original coupled RMAB under the same kernels, hence independent of learning error. If $h(N)\neq 0$ (e.g., finite $N$, no global-attractor structure), the linear term dominates for large $T$, nullifying the sublinear part. The algorithm may learn quickly under drift, but if Whittle is structurally sub-optimal in the domain, guarantees are ultimately bottlenecked by $h(N)T$. Hence, a discussion is needed when $h(N)\approx 0$.

2. Computational burden of inner ``optimistic kernel'' maximization (Eq. 12):
Per arm/episode, solves Eq. 12-13 over a high-dimensional confidence polytope $\mathcal B_t^{(n)}$. Outside special 1-D structure, there is no closed form, and the paper itself flags computational intensity. Therefore, without structure, per-episode overhead can limit scalability for large $|S|$ or $N$, creating a gap between theory and deployment.



3. Evaluation scope is narrow:} Experiments run for $T=50$ episodes and rely on domain simplifications to solve Eq.~(12). This under-stresses the asymptotic trade-offs exposed by the theory and does not report wall-clock scaling. Stronger ablations (vary $J_n$, $w_n$, misspecify $S_0$) and runtime plots would better substantiate the method's practicality.

**Questions:**

Q1: Can you empirically probe regimes where $h(N)$ is non-negligible (finite $N$, no global-attractor), showing the linear term's impact?

Q2: How sensitive is performance to misspecified $S_0(s,a)$? Could one learn $S_0$ online with penalties?

Q3: Any simple exploration schedule to ensure $P_{\min}$ does not collapse when $M\ll N\$?


Q4: Please include runtime scaling (vs. $|S|,N$) for Eq.~(12) in both domains.

---

> ### Author Response · Authors · 2025-11-21
> **Response to Weakness and Questions of Reviewer bw55**
>
> We thank the reviewer for the constructive feedback. Below, we address the specific questions raised. We have updated the paper accordingly.
>
> *Weakness*
>
> **Term $h(N)T$:** We agree with reviewer that if $h(N)$, is non-zero, the resulting additive term will dominate the sublinear regret for large $T$. However, this term represents an inherent limitation in RMAB domain, independent of the learning mechanism. Since Whittle (1988), it has been understood that Whittle policy can be suboptimal even under stationary conditions. True optimal policy is PSPACE-hard (Papadimitriou & Tsitsiklis (1994)). Our work's primary focus is not to solve this long-standing, fundamental problem of optimal RMAB policy design. Instead, we focus on obtaining the best possible Whittle policy when the transitions are non-stationary, which is a distinct challenge. In the paper, we have mentioned briefly in Remark 1, when $h(N)\approx 0$. For numerical results, we use the best Whittle policy to demonstrate the regret. This is because it is not possible to get the optimal policy even if everything is known exactly. This is a common practice in RMAB literature (Wang et al. (2023)).
>
> **Complexity of Eq 12:** In Appendix A.6, we have now added detailed computational complexity analysis. In many practical settings, including wireless scheduling and 1D bandits, we can get closed form solutions for (12), making the bilinear optimization very efficient. In settings where we do not have closed form solutions, we can use extended value iteration (EVI) algorithm [R1]. In EVI of each arm, in every iteration, we visit every state to update the value function. The complexity to update the value function for each state is $O(|\mathcal S|^2 |\mathcal A|)$, whereas the complexity of value iteration algorithm is $O(|\mathcal S| |\mathcal A|)$, where $|\mathcal S|$ and $|\mathcal A|$ are total number of states and actions, respectively. The extra computation, we need is to solve the linear problem $$ \max_{P_{n, t} \in B_t^{(n)}}\sum_{s^{'}\in \mathcal S} P_{n, t}(s'|s, a) V_{n, t, \lambda}(s'), $$ which takes $O(|\mathcal S|)$ time [R1]. In our simulation, one iteration in EVI algorithm takes 0.037 sec and one iteration using closed form solution takes 0.006 sec.
>
> [R1] Auer et. al., Near-optimal regret bounds for reinforcement learning, NeurIPS, 08.
>
> **narrow Evaluation scope**  We have incorporated new simulation results in Table 1 and Figure 1 for the wireless scheduling problem using a real dataset (Reddy et al. (2025)) that contains traces of measured signal strength across indoor and outdoor settings, leading to non-stationary behavior. In Appendix A.5, we plot the success probability values, which clearly demonstrate the time-varying nature of the dataset. Our added numerical results directly show that our algorithm outperforms all prior baselines in the real-world wireless dataset. We also added Table 2, which compares the results for different values of $J_n$ and includes scenarios both with and without the knowledge of $V_n$.
>
> *Questions*
>
> Q1 While we agree that evaluating the impact of $h(N)$ in settings where the Whittle is sub-optimal would be interesting, it's quite challenging to do in practice. First, for a lot of applications that utilize the Whittle index, it tends to be asymptotically optimal. Second, calculating the true gap $h(N)$ requires comparing the Whittle policy's performance against the optimal policy. Determining the optimal policy is generally PSPACE-hard, making such a comparison computationally intractable even when $N$ is not too large. We acknowledge that the term's impact is an inherent limitation of the Whittle index approach itself, a problem separate from the non-stationarity challenge, and outside the scope of our current work.
>
> Q2 Our algorithm and theoretical results hold without explicit knowledge of the sparsity. Sparsity helps us to reduce the computation time. If sparsity is unknown, we can simply ignore the sparsity and consider the full state space as the set of possible next transitions. As algorithm explores and time horizon increases, the true transitions will still be learned effectively, but with larger value of time steps in one episode and more computation time.
>
> Q3 Two effective schedules are: Using a larger time horizon in each episode which provides more opportunity for exploration and can increase the probability to visit every state action, i.e., $P_{min}$. Another way is to implement an $\eta$-random policy. Employing an $\eta$-random schedule guarantees persistent exploration. With a small, positive probability ($\eta$), we execute a uniform random policy; otherwise (with probability $1-\eta$), we execute the learned Whittle policy. Practically, for our numerical results, we don't need to use the $\eta$-random schedule since the episode lengths we consider are long enough.
>
> Q4 In response to weakness, we have discussed runtime for (12).
>
> Please let us know if you have any further questions.

---

### Official Review · Reviewer_SVWM · 2025-11-03

**Soundness:** 3
**Presentation:** 3
**Contribution:** 3
**Rating:** 6
**Confidence:** 4

**Summary:**

This paper studies online learning in restless multi-armed bandits (RMABs) where transition kernels are unknown and non-stationary. The authors propose a Sliding-Window Online Whittle (SW-Whittle) algorithm. This method integrates Whittle index computation with a sliding-window and bandit-over-bandit framework to adapt to changing dynamics. They prove a dynamic regret bound and validate their method on two simulated RMAB problems (wireless scheduling and a one-dimensional bandit), showing improved performance over UCWhittle and other baselines.

**Strengths:**

+ Clear motivation for addressing non-stationary RMABs.
+ Novel algorithm design combining Whittle index, UCB, and sliding-window learning.
+ Strong theoretical guarantees with explicit regret scaling.
+ Experiments show consistent improvement over prior algorithms.
+ Discussion of sparsity in transition kernels adds realistic motivation

**Weaknesses:**

- The empirical evaluation is limited to small synthetic environments. Real-world data or larger-scale experiments would improve credibility.
- The algorithm’s computational complexity (especially in solving the bilinear optimization in Eq. 12) is acknowledged but not analyzed quantitatively.
- Some assumptions, such as known sparsity structure (assuming indexability is given), may not always hold in practice.

**Questions:**

1. How sensitive is performance to the quantization level in the bandit-over-bandit step?
2. Could the method extend to continuous or infinite state spaces via function approximation?
3. Can the approach handle partial observability or contextual features?
4. How does computation time scale with the number of arms and states?

---

> ### Author Response · Authors · 2025-11-21
> **Response to Weakness and Questions of Reviewer SVWM**
>
> We thank the reviewer for the constructive feedback. Below, we address the specific questions raised. We have updated the paper accordingly.
>
> **Weakness**
>
>  **Real World Data** We have incorporated new simulation results in Table 1 and Figure 1 for the wireless scheduling problem using a real dataset (Reddy et al. (2025)) that contains traces of measured signal strength across indoor and outdoor settings, leading to non-stationary behavior. In Appendix A.5, we plot the success probability values, which clearly demonstrate the time-varying nature of the dataset. Our added numerical results directly show that our algorithm outperforms all prior baselines in the real-world wireless dataset.
>
> **computational complexity** In Appendix A.6, we have now added detailed computational complexity analysis. In many practical settings, including wireless schedule and 1D bandits, we can get closed form solutions for Eq. 12, making the bilinear optimization very efficient. In settings where we do not have closed form solutions, we can use extended value iteration algorithm [R1]. In the extended value iteration algorithm of each arm, in every iteration, we visit every state $s\in \mathcal S$ to update the value function. The complexity to update the value function for each $s\in \mathcal S$ is $O(|\mathcal S|^2 |\mathcal A|)$, whereas the complexity of value iteration algorithm is $O(|\mathcal S| |\mathcal A|)$, where $|\mathcal S|$ and $|\mathcal A|$ are total number of states and actions, respectively. The extra computation, we need is to solve the linear problem $$ \max_{P_{n, t} \in B_t^{(n)}}\sum_{s^{'}\in \mathcal S} P_{n, t}(s'|s, a) V_{n, t, \lambda}(s'), $$ which takes $O(|\mathcal S|)$ time [R1]. In our simulation, one iteration
> in extended value iteration algorithm takes 0.037 sec and one iteration using closed form solution takes 0.006 sec.
>
> [R1] Auer et. al., ``Near-optimal regret bounds for reinforcement learning" Advances in neural information processing systems, 2008.
>
> **Assumptions** Our algorithm and results hold without any assumption of sparsity. The sparsity helps us find computationally efficient solutions to Eq. 12. The complexity to update the value function for each $s\in \mathcal S$ reduces to $O((|\mathcal S-\mathcal S_0(s, a)|)^2 |\mathcal A|)$, if we know the sparsity information $\mathcal S_0(s, a)$. Further, if we know sparsity (even approximately), we can use this information to learn faster by reducing exploration for certain transitions. Indexabilty is a typical assumption for RMABs and usually comes from domain knowledge and structural information of the RMABs. Prior works have proved the indexabilty for RMABs in a wide variety of applications. For example, it holds for both the wireless scheduling and one dimensional bandit settings discussed in the paper.
>
> **Questions**
> **Sensitivity of quantization level** Theorem 2 directly quantifies the fundamental trade-off introduced by the quantization level in the BoB approach: increasing $J_n$ improves the accuracy of tracking the variation budget ($V_n$) but concurrently slows down the BoB learning. The effectiveness of the approach is empirically evaluated in the new Table 2 in the Appendix, which shows that performance is robust and not highly sensitive to the exact value of $J_n$, provided a sufficiently large level is chosen.
>
> **Continuous or Infinite State Spaces** Our method can be extended to continuous or infinite state spaces via function approximation, such as by integrating tools like DQN into Equation (12). However regret analysis with function approximation becomes much more challenging. This is a promising direction for future work.
>
> **Partial Observability or Contextual Features** Crucially, RMABs are different from classical multi-armed bandits, since each arm itself is an MDP and its state can effectively model contextual features (as long as context is finite dimensional). Thus, incorporating context is straightforward and almost by design for our problem. On the other hand, partial observability is much more challenging to handle for RMABs (both with and without learning). This is also a promising direction for future work.
>
> **Computation Time** We have added detailed computational complexity analysis in Appendix A.6. In the beginning of every episode, the predicted transition kernels is computed for all arms by Eq. (12). The complexity to update the value function for each state in solving (12) is $O(|\mathcal S|^2 |\mathcal A|)$, Then, we compute Whittle index for all states. We can use closed form equation or use bisection method to solve (7). With tolerance $\text{tol}$, upper bound $u$ and lower bound $l$, for each state, it can take at least $O(\mathrm{log}2((u-l)/\text{tol}))$. Next, at every time step of each episode, we sort $N$ arms using index and select $M$ arms with highest indices which take $O(N\mathrm{log}N)$ time.
>
> Please let us know if you have any further questions.

---

### Official Review · Reviewer_p3aE · 2025-11-10

**Soundness:** 3
**Presentation:** 3
**Contribution:** 2
**Rating:** 4
**Confidence:** 3

**Summary:**

This paper addresses the challenge of resource allocation in restless multi-armed bandits (RMABs) where transition kernels are unknown and non-stationary. The authors propose the Sliding-Window Online Whittle (SW-Whittle) policy, which adapts to time-varying dynamics by using sliding windows for kernel estimation, upper-confidence bounds, and a Bandit-over-Bandit framework to handle unknown variation budgets. Key contributions include the algorithm design, a dynamic regret bound of $\tilde{O}(T^{2/3} \tilde{V}^{1/3} + T^{4/5})$, and numerical simulations demonstrating superior performance over baselines like UCWhittle and WIQL.

**Strengths:**

- **First work to tackle Non-stationarity in RMABs**: This is the first work to provide dynamic regret bounds for online learning of Whittle indices in non-stationary RMABs, extending stationary methods like UCWhittle (Wang et al., 2023) and addressing a gap in the literature where prior non-stationary RMAB works.

- **Rigorous theoretical analysis**: The dynamic regret bound $\tilde{O}(T^{2/3} \tilde{V}^{1/3} + T^{4/5})$ is well-derived, improving on stationary regrets and directly analyzing the main problem rather than Lagrangian relaxations, as in Wang et al. (2023).

- **Novelty on using Bandit-over-bandit approach**: The algorithm handles unknown variation budgets via Bandit-over-Bandit and exploits kernel sparsity, making it adaptable to real-world sparsity in applications like Age of Information in wireless scheduling.

- **Clear presentation and detailed literature review**: The paper is well-structured, with a detailed literature review and algorithm descriptions, and proofs in appendices.

**Weaknesses:**

- **Limited experimental validation with synthetic data on a toy setup**: While simulations show lower regret, they rely on synthetic environments; real-world datasets for RMABs exist (e.g., ARMMAN dataset), which could provide more convincing evidence of applicability. This ties into the lack of mention of specific real-world datasets or problems with this non-stationarity modeling, despite bandits' alignment with practical issues such as user preferences drifting over time.

- **Suboptimality gap of Lagrangian relaxation with non-stationarity**: The suboptimality from Lagrangian relaxation (with gap $h(N) \to 0$ as $N \to \infty$) and its interaction with non-stationarity is not thoroughly analyzed/explained. Changing kernels could shift the "goalpost" for large $N$ approximations, yet this is not explicitly addressed.

- **Implications of the non-stationarity model**: The model bounds per-episode changes by $V_n/T$, implying total variation $V_n$, where for sublinear regret $V_n = o(T)$ and average non-stationarity diminishes as $T \to \infty$; this assumes environments that stabilize over time. Was this intentional? The lack of real-world examples tied with the modeling assumptions is making it hard to justify what can be allowed and what is a stretch.

**Questions:**

- **Choice of $w_n$**: The window size $w_n = \lceil (T/V_n)^{2/3} \rceil$ appears constructed specifically to achieve the regret bound. Any more discussion on where and why this would be the correct choice?

- **Bandit-over-bandit approach**: While this increases the novelty of the overall work, this hasn't been discussed much and not touched much in experiments as well. Can you expand on the results and insights surrounding it more?

- **Impact of Sparsity**: Apologies if I missed this, but after being touched upon, the role of sparsity and its impact on the final results and experiments seem to be undermined. Can you expand on this as well?

- Is the $T^{4/5}$ term in the regret bound necessarily optimal? What is the scope of improvement? Additionally, what real-world problems fit the $V_n/T$ diminishing non-stationarity model?

---

> ### Author Response · Authors · 2025-11-21
> **Response to Weakness and Questions of Reviewer p3aE**
>
> We thank the reviewer for the constructive feedback. Below, we address the specific questions raised. We have updated the paper accordingly.
>
> **Limited experimental validation**
> We have incorporated new simulation results in Table 1 and Figure 1 for the wireless scheduling problem using a real dataset (Reddy et al. (2025)) that contains traces of measured signal strength across indoor and outdoor settings, leading to non-stationary behavior. In Appendix A.5, we plot the success probability values, which clearly demonstrate the time-varying nature of the dataset. Our added numerical results directly show that our algorithm outperforms all prior baselines in the real-world wireless dataset. For ARMMAN dataset, we reached out to one of the authors of (Wang et al. (2023)). It turns out that the dataset itself is not available to the public due to the privacy concerns with sharing health data of participants.
>
> **Suboptimality gap of Lagrangian with non-stationarity**
> The crucial point is that for any predicted transition probability, the term $h(N)\to 0$ as $N\to \infty$. The $h$ function measures the gap between an optimal policy and a Whittle policy in a stationary setting, and is typically small and converging to zero for large classes of RMABs. In other words, this result is a property of the large-scale structure of RMAB problems and holds true regardless of the non-stationary transitions. Therefore, the asymptotic analysis (as $N \to \infty$) does not have any interaction with the non-stationarity. The asymptotic property typically holds when the indexability and the global attractor conditions are satisfied. While indexability have been established for many setups, the global attractor is often used as an assumption. For specific problems, such as wireless scheduling minimization, [R1] showed that the suboptimality gap, $h(N)$, vanishes without using any assumption.
>
> [R1] S. Kriouile et. al., On the global optimality of Whittle’s index policy for minimizing the age of information.
>
> **Implications of the non-stationarity**
> The assumption of sublinear total variation for the transition probabilities does not necessarily imply that the environment stabilizes over time. Consider the example of transition kernels that change by a fixed amount every $\sqrt{T}$ episodes. While the total variation is sublinear in $T$, the kernels themselves are not stabilizing over time. Using sublinear total variation is standard practice in dynamic regret literature as it's a necessary condition to achieve sublinear dynamic regret. Within online learning and reinforcement learning literature [R2] which show that linear growth of variation budgets *must* imply linear dynamic regret, regardless of algorithm choice.
>
> [R2] Besbes et al. Stochastic Multi-Armed-Bandit Problem with Non-stationary Rewards, NeurIPS, 2014
>
> Questions:
> **Choice of $w_n$**  It's indeed true that the window size is constructed specifically to achieve the best regret bound. In particular, it's one of the main parameters that influences regret as analyzed in Theorem 1. Following this analysis, in Theorem 2, we provide the precise choice of $w_n$ that yields the best regret bound under our approach. Intuitively, if $V_n$ is small, so we should use longer window sizes and get better estimates. On the other hand, if $V_n$ is large, using too much history can lead to large errors in kernel estimation.
>
> **Bandit-over-bandit approach**The BoB approach is needed because we assume that $V_n$ is unknown. Our main results demonstrate the impact of BoB on the regret bound. Specifically, the third term in the regret bound presented in Theorem 1 is directly related to the learning error of the  BoB. Furthermore, in Theorem 2, the influence of the BoB on the overall regret is quantified through the parameter $J_n$. In the Appendix, we have added Table 2 comparing the performance of our policy with known and unknown variation budgets. This new result shows that the BoB is effective.
>
> **Impact of Sparsity** Our regret analysis results are general and do not need a sparsity assumption. However, sparsity helps greatly in simplifying the computational complexity of our algorithm. We have made this clear after the definition and Appendix A.6 discusses complexity.
>
> **$T^{4/5}$ Term** The term in the regret bound specifically arises for the scenario where the variation budget is unknown: for known variation budget, the term vanishes and we achieve $T^{2/3}$ regret bound, which can be the best possible regret [R2].  This $T^{4/5}$ term arises directly from our particular choice for the set of possible budgets used in BoB. A different design could yield a different term. To establish whether our bound is truly optimal, a further analysis of regret lower bound is required. Proving such lower bounds is a complex challenge and an open area for RMAB. In Section 7, it mentioned as a future direction.
>
> Please let us know if you have any further questions.

---

### Author Response · Authors · 2025-12-04
**Rebuttal Summary and Manuscript Changes**

Dear Area Chairs:

Thank you for handling our submission. We addressed reviewers' concerns and updated our paper. Below, we summarize our main revisions.

**New Simulation Results:** We have incorporated new simulation results in Table 1-2 and Figure 1 for the wireless scheduling problem using a real dataset (Reddy et al., 2025) that contains traces of measured signal strength across indoor and outdoor settings, leading to non-stationary behavior. In Appendix A.5, we plot the success probability values in Figure 2 and Variations in Figure 3, which clearly demonstrate the dataset's time-varying nature. Our added numerical results directly show that our Sliding-Window Online Whittle (SW-Whittle) policy outperforms all prior baselines in the real-world wireless dataset.

**Computational Complexity:** We have incorporated a detailed computational complexity analysis into Appendix A.6 to address the reviewers' concerns regarding the complexity of solving Equation (12). The solution to this equation can be obtained either via a closed-form expression or through the Extended Value Iteration (EVI) algorithm. We clarify that Equation (12) is solved once per episode, not at every individual time step. When a closed-form solution is unavailable, the EVI algorithm is employed, where each iteration requires $O(|\mathcal S|^2 |\mathcal A|)$ time to update the value function for every state $s \in \mathcal S$.

**Sparsity:** Our proposed algorithm can leverage sparsity information to reduce its complexity, as described in Appendix A.6. Further, if we know sparsity (even approximately), we can use this information to learn faster by reducing exploration for certain transitions. Even in the absence of sparsity information, our results hold, and the proposed algorithm guarantees sublinear dynamic regret.

**Regret Bound $O(T^{2/3}V^{1/3}+T^{4/5})$:** The $T^{4/5}$ term in the regret bound specifically arises for the scenario where the variation budget is unknown: for known total variation budget, the term vanishes and we achieve $O(T^{2/3})$ regret bound, which is comparable to other MDP works with non-stationary transition kernels (Ortner et al. (2020)). This $T^{4/5}$ term arises directly from our particular choice for the set of possible variation budgets used in the Bandit over Bandit (BoB) approach. A different design for this set could yield a different term. To establish whether our bound is truly optimal, a further analysis of the regret lower bound for any general algorithm is required. Proving such lower bounds in non-stationary MDP settings is a complex challenge and an open area of exploration. This is a promising direction for future work for us, as mentioned in Section 7.

We also want to mention that, according to our earlier conversation with the Area chairs, we have avoided comparing our work to the paper "Non-Stationary Restless Multi-Armed Bandits with Provable Guarantee," as initially requested by Reviewer KPF4. This is to avoid violating double blind guidelines.

The new simulation results and analysis significantly strengthen our paper's main contributions and theoretical soundness as a unique solution to the RMAB problem with non-stationary transition kernels.

---

### Meta-Review · Area_Chair_pC1R · 2026-01-02

**Summary:**

This paper studies restless multi-armed bandits (RMABs) where per-arm transitions are unknown and the transition kernels drift over time, while the decision maker activates $M$ of $N$ arms each time. Method-wise, this paper proposes an approach named SW-Whittle, i.e., a Sliding-Window Online Whittle. Basically, SW-Whittle estimates transitions from a recent window, based on which, it builds an optimistic transition model inside a confidence set, and then computes the optimistic Whittle indices via a bilinear optimization to select the top $M$ indices. To deal with unknown non-stationary, SW-Whittle adds a bandit-over-bandit (BoB) tuning of the window lengths using quantized candidate variation levels. This paper proves that SW-Whittle achieves a dynamic regret for large RMABs.

The problem setting, i.e., non-stationary RMABs, is practical and interesting. However, the proposed approach is fairly general and largely follows the same blueprint as existing methods for stationary settings, i.e., building optimistic transition models within confidence sets and then computing Whittle indices. Arguably, the most notable algorithmic component is the EXP3-style procedure for adaptively tuning the window length.

On the theory side, the dynamic-regret guarantee is understandably limited: it does not provide an optimality guarantee, depends on specific design choices, and is primarily meaningful in large-system regimes. In particular, the regret analysis decomposes regret in a way that naturally yields an additional term $h(N)T$, where $h(N)$ is the approximation gap between the Whittle policy and the true RMAB optimum. This term vanishes only asymptotically under additional conditions. Consequently, for many practical systems with moderate $N$, $h(N)T$ can dominate. Note that the RMAB/Whittle model is intended for systems of any size, not only "large-scale"settings. Thus, the theoretical guarantee is mainly informative in the large-system regime. In addition, the regret bound contains a $T^{4/5}$ term that depends on the discretized set of budgets used in BoB, and alternative designs could change this term (potentially making it closer to linear in $T$), and the current analysis does not establish any optimality for this dependence.

Moreover, one reviewer requested comparisons against stationary-regret bounds in prior work, and the authors claimed that "(Wang et al., 2023) is the only paper that provides stationary regret bounds for RMAB." This claim is clearly incorrect, as there are multiple recent works (at least from 2019 to 2025) that provide regret guarantees for stationary RMABs.

Some reviewers also raised concerns about the assumptions used in the analysis. As is conventional, the Whittle asymptotic-optimality machinery, e.g., indexability plus a global attractor condition, is invoked, and the learning analysis further relies on technical conditions such as minimum visitation and probability surrogates, e.g., a $P_{\min}$ term appearing in the window-length optimization. These assumptions should be more explicitly tested and discussed in the experiments and limitations.

Regarding proof novelty, the technical contributions appear to be primarily adaptations rather than fundamentally new tools. The non-stationary confidence-set construction, i.e., combining a statistical estimation term with a drift term proportional to the window length, is standard and widely used in analyses of non-stationary MDPs and RL. Thus, the paper does not introduce genuinely new proof techniques. Instead, its contribution is better viewed as a new combination/adaptation of existing ideas to the non-stationary RMAB setting. Specifically, pushing these tools through RMAB structure and Whittle-index computation to obtain a clean dynamic-regret scaling.

Empirically, the authors added additional results during rebuttal. However, the evaluation still covers only a limited set of RMAB benchmarks. While the paper claims consistent improvements across non-stationary environments, this claim is not fully supported by the current experimental scope. The authors should consider additional case studies of RMAB applications, larger $N$, sensitivity analyses across drift patterns, runtime comparisons, and stronger comprehensive baselines.

**Reviewer Concerns:**

The rebuttal clarified the concerns on the regret decomposition and where the extra Whittle-gap term comes from. This question has been asked by several reviewers and the authors explained the regret decomposition in the rebuttal and how they obtained the regret bound by setting different parameters, window sizes, etc.

Reviewers (at least two) may still be skeptical about the technical novelty of this work. Technically, the proof strategy largely follows known non-stationary RL patterns, i.e., sliding-window confidence with drift term, and UCWhittle-style optimistic bilinear planning.  The novelty is mainly the RMAB and Whittle integration and bounding. Some reviewers may still view this as "incremental," unless the final version better highlights what is genuinely RMAB-specific in the analysis.

In addition, the $h(N)T$ term can dominate unless $N$ is large and the Whittle asymptotic conditions apply.  This should be more explicitly positioned as a large-system guarantee, with stronger empirical evidence for moderate $N$.

On the empirical side, even if additional results are provided during the rebuttal, reviewers may still want broader non-stationarity patterns, larger-scale settings, and runtime comparisons, since practicality is a key selling point of Whittle-based learning.

**Reviewer Scores:**

Reviewer SVWM's comments are addressed mostly in terms of claiming as future works. One remaining concern is about the computational complexity, which can be very costly if the system is large in the sense of having many states. The reviewer's score may be unchanged.

Reviewers p3aE, bw55 and KPF4 raised several technical questions regarding the regret bounds, proofs, etc. The authors respond in the rebuttal, however, many concerns may still remain as the regret bound is not guaranteed to be optimal and highly depend on the parameter selections. Their scores may remain unchanged.

---

### Decision · Program_Chairs · 2026-01-26

Reject